# What Do VLMs See? Benchmarking Vision-Language Models on Ambiguous Images

## Abstract

Vision–language models (VLMs) have demonstrated remarkable capabilities in visual recognition and reasoning, in some cases even surpassing human performance on standard benchmarks. However, it remains largely unexplored whether VLMs possess higher-order aspects of human perception, such as abstract interpretation and the capacity to manage cognitive ambiguity, and to what extent. In this paper, we introduce **AmbiBench**, a benchmark designed to systematically evaluate how VLMs perceive and reason about ambiguous images relative to human interpretations. AmbiBench comprises 2,250 ambiguous images spanning nine categories, including object-level, scene-level, and a newly introduced mixed-ambiguity class, paired with 2,723 carefully constructed visual question–answer pairs. Evaluation of 12 state-of-the-art VLMs reveals substantial limitations: in five categories, models achieve less than half of human accuracy, and on mixed-ambiguity images, most collapse to near-zero performance. Our study shows that humans flexibly navigate multiple interpretations, shifting between global and local perspectives, whereas VLMs largely rely on dominant features and exhibit restricted perception and reasoning under ambiguity. We further probe the existence of perceptual-switch heads—attention mechanisms that may underlie cognitive ambiguity—using bistable images. AmbiBench exposes critical gaps in current VLMs' capacity to handle perceptual ambiguity and establishes a foundation for developing models with more human-aligned interpretive and reasoning abilities.

## 1 Introduction

Large vision–language models (VLMs) (Zhu et al., 2023; Liu et al., 2023; OpenAI, 2024; Lu et al., 2024a) have advanced rapidly in recent years, demonstrating strong performance across tasks such as image captioning (Zeng et al., 2024), visual understanding (Yu et al., 2023), and multimodal reasoning (Wang et al., 2024). While VLMs now approach human-level performance on standard benchmarks for fundamental visual interpretation, their alignment with higher-order aspects of human cognition (Bialystok & Shapero, 2005; Wiseman et al., 2011), including cognitive flexibility, creativity, imagination, and the ability to handle perceptual ambiguity, remains largely unexplored.

Ambiguous images, defined as visual stimuli that are either deliberately constructed or naturally occurring and allow multiple valid interpretations, present a distinctive challenge for both human perception and artificial intelligence (AI) systems (Long & Toppino, 2004; Leopold & Logothetis, 1999; Sterzer et al., 2009)(Ullman, 2024). In psychology (von Helmholtz, 2001) and neuroscience, such stimuli have long been used to probe higher-order perceptual processes, offering insight into how humans resolve ambiguity, entertain competing hypotheses, and adapt interpretations over time. Despite their significance, a comprehensive benchmark for assessing how VLMs process and reason under perceptual ambiguity remains lacking, primarily due to the following challenges:

- **Difficulty in obtaining diverse and high-quality ambiguous images.** Ambiguous stimuli span a broad range of visual phenomena, including color-based illusions (Guan et al., 2024), bistable perception (Panagopoulou et al., 2024), motion illusions, geometric distortions. Constructing a dataset that is both sufficiently large and representative of this diversity, while also maintaining high visual quality, remains a major challenge.

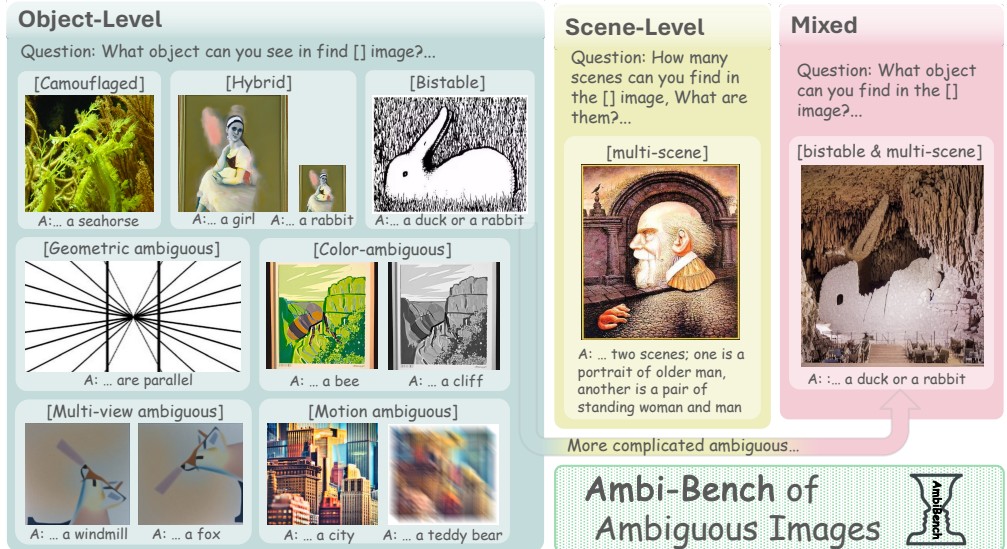

Figure 1: Examples of different ambiguous categories in the proposed AmbiBench.

- **Lack of standardized evaluation protocols.** Unlike conventional vision tasks with clearly defined ground truth, ambiguous images permit multiple valid interpretations. This makes it difficult to design evaluation tasks and metrics that reliably measure how well AI models capture human-like perceptual flexibility and reasoning under ambiguity.

In this paper, we introduce *AmbiBench*, a benchmark specifically designed to evaluate the perceptual and interpretive capacities of VLMs when confronted with ambiguous visual stimuli. AmbiBench spans nine categories of ambiguous data: object-level ambiguity (camouflaged (Le et al., 2019), bistable (Panagopoulou et al., 2024), hybrid (Oliva et al., 2006), color-based (Lafer-Sousa et al., 2015), multi-view (Liu et al., 2023), geometric (Guan et al., 2024), and motion-related (Taranu et al., 2019; Geng et al., 2024a)), scene-level ambiguity (multi-scene), and a newly proposed category, mixed ambiguity, in which two forms of ambiguity coexist within a single image (examples in Fig. 1; detail description in the Appendix A.1). The images are drawn from natural photography, artistic renderings, and AI-generated content. After careful filtering, AmbiBench contains 2,250 ambiguous images paired with 2,723 question–answer items, encompassing open-ended questions, multiple-choice questions, ambiguous grounding, and local region description tasks. This design enables comprehensive evaluation of both global and local perceptual processing, as well as cognitive control, under conditions of visual ambiguity.

We systematically evaluate 12 state-of-the-art VLMs on AmbiBench and find that, across five categories, models achieve less than half of human-level accuracy. Performance collapses to near-zero on highly ambiguous stimuli, particularly those involving mixed ambiguities, underscoring the limited imaginative and cognitive flexibility of current models in handling complex perceptual ambiguity. Whereas humans readily entertain multiple plausible interpretations, flexibly shifting between global and local perspectives, VLMs tend to default to dominant visual features or contextually biased labels, revealing limited capacity for flexible reasoning under ambiguity. We further examine whether VLMs contain internal mechanisms analogous to human perceptual switching in bistable vision, where specific neural activations support alternation between interpretations.

Our contributions can be summarized as follows:

- We present **AmbiBench**, the first benchmark systematically curated to evaluate ambiguous image perception in VLMs. It spans nine categories of visual ambiguity, including a newly introduced mixed category, and covers text, image, and video modalities.
- We evaluate 12 state-of-the-art VLMs on open-ended, multiple-choice, ambiguous localization, and local region description tasks, directly comparing model outputs with human responses and uncovering substantial differences in handling ambiguity.

- We probe whether VLMs contain specialized mechanisms for perceptual switching in bistable and multi-scene images and identify perceptual-switch heads. We further show that positive interventions on these heads can improve model performance.

## 2    RELATED WORK

**Visual Ambiguity.**    Visual Ambiguity, often referred to as a visual or optical illusion, broadly describes perceptual phenomena in which a single stimulus can elicit multiple, sometimes conflicting, interpretations. Cognitive psychology has long examined such phenomena, including bistable perception (Devia et al., 2022), brightness and color illusions (Lafer-Sousa et al., 2015), and geometric and motion illusions (Westheimer, 2008; Taranu et al., 2019), to understand how the visual system resolves ambiguity. Beyond classical illusions, hybrid images introduce spatial ambiguity by producing different interpretations at varying viewing distances (Oliva et al., 2006). Camouflaged-object datasets highlight feature-based ambiguity (Le et al., 2019), while multi-view illusions change appearance under transformations such as rotation or flipping (Geng et al., 2024b; Xu et al., 2025).

With the rapid advancement of generative AI, recent studies (Geng et al., 2024b; Gao et al., 2025b; Geng et al., 2024a; Xu et al., 2025; Burgert et al., 2024) have demonstrated the synthesis of diverse ambiguous images, including multi-view, color, motion, and hybrid ambiguities. Building on these categories, we also consider multi-scene images that embed layered or overlapping scenes. Most importantly, we introduce a new mixed category that combines bistable and multi-scene ambiguities, yielding more complex and challenging stimuli for assessing the perceptual flexibility of VLMs.

**Ambiguity Benchmarks for VLMs.**    Recent work has begun to evaluate the perceptual abilities of VLMs on ambiguous images. However, most existing benchmarks are restricted to a narrow set of categories, such as bistable images (Newen et al., 2025; Panagopoulou et al., 2024), color and geometric illusions (Guan et al., 2024; Zhang et al., 2023), or multi-scene cases with a single hidden element (Li et al., 2025). In addition to this limited diversity, current benchmarks (Shahgir et al., 2024) are small in scale and constrained in task design, preventing a comprehensive evaluation of VLMs. In contrast, our proposed **AmbiBench** comprises more than 2,000 images (including videos for motion ambiguity) across nine categories with varying forms of ambiguity. We further design four task types, each paired with carefully constructed QA items, to enable systematic assessment of VLMs under ambiguous perception. It is important to distinguish ambiguous images from so-called weird or counterfactual images, which are often AI-generated and designed to break common sense (Bitton-Guetta et al., 2023). Whereas counterfactual images depict unrealistic or nonsensical content, ambiguous images are grounded in perceptual conflict and support multiple plausible interpretations. The "test-eye" evaluations (Gao et al., 2025a), which target low-level visual features and object-centric perception, also differ from the perceptual ambiguity examined in this work.

## 3    AMBIBENCH

### 3.1    OVERVIEW OF AMBIBENCH

We present **AmbiBench**, a benchmark designed to comprehensively evaluate how VLMs align with higher-order aspects of human perception, including abstract interpretation and the ability to manage cognitive ambiguity. As shown in Fig. 2, AmbiBench covers three major types of ambiguity: object-level, scene-level, and mixed. The object-level type includes seven categories: camouflaged, bistable, hybrid, color, multi-view, geometric, and motion. For each image, we construct carefully designed question–answer pairs that assess both fundamental understanding of visual illusions through open-ended and multiple-choice formats, as well as more advanced abilities such as ambiguous localization and local region description. In total, AmbiBench provides 2,723 QA pairs, enabling systematic and multi-faceted evaluation.

### 3.2    DATA CURATION

The central challenge in constructing an ambiguous image benchmark lies in acquiring sufficient high-quality samples. Unlike conventional vision datasets, ambiguous images require deliberate design or artistic creation rather than straightforward web crawling. To address this, we combined

images from online repositories and public datasets with additional samples generated by diffusion models using carefully crafted prompts, thereby ensuring both scale and diversity.

### 3.2.1 OBJECT-LEVEL CATEGORIES

**Camouflaged.** Camouflaged images, a form of feature-grounded ambiguity, depict natural scenes in which objects are intentionally concealed within complex backgrounds, making them difficult to detect. By leveraging perceptual blending to obscure object boundaries, such images pose significant challenges for both human observers and machine perception. From the CAMO dataset (Le et al., 2019), we filtered images with pixel-level annotations and retained 1,140 high-quality samples. Gemini (Comanici et al., 2025) was used to generate initial camouflaged object labels—primarily animals—which were grouped into 69 categories. Annotators then manually corrected and aligned these labels to a set of 69 defined categories, ensuring that each image contained only a single camouflaged class.

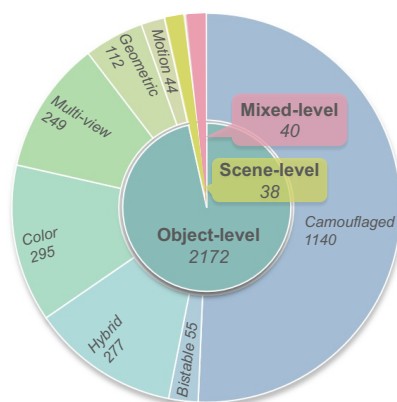

Figure 2: The distribution of ambiguity categories in AmbiBench.

**Bistable.** Bistable images (Leopold & Logothetis, 1999) are visual stimuli that afford two distinct interpretations, although only one can be perceived at a time. We curated 55 examples from diverse online sources and prior literature (Newen et al., 2025), including classic cases such as the Duck–Rabbit and the Young–Old Woman (Panagopoulou et al., 2024).

**Hybrid.** Hybrid images (Oliva et al., 2006) exhibit distance-dependent perceptual ambiguity, appearing differently depending on viewing distance. They typically embed multiple scenes or objects at distinct spatial frequencies, creating layered interpretations. Such images can be generated by decomposing an image into frequency subbands and combining them with different prompts in diffusion models. Following Geng et al. (2024a), we adopt a two-component decomposition $\mathbf{x} = (\mathbf{x} - G_\sigma(\mathbf{x})) + G_\sigma(\mathbf{x})$, where $G_\sigma$ is a Gaussian blur with standard deviation $\sigma$, and $\mathbf{x} - G_\sigma(\mathbf{x})$ serves as the high-pass component. Using this approach, we synthesize hybrid images by assigning two prompts to the two frequency bands. In addition to synthetic cases, hybrids can also be derived from real images and extended to triple hybrids (near, far, very far). By combining online sources, the gallery from Geng et al. (2024a), and our diffusion-generated samples, we curated 277 hybrid images, including 7 triple hybrids.

**Color.** Color-based stimuli can produce multiple interpretations depending on how chromatic information is perceived. In AmbiBench, we focus on color hybrid images, which exhibit different appearances when viewed in grayscale versus color. To expand this category, we generate additional samples using a diffusion model with a color-space decomposition following Geng et al. (2024a):

$$f_{\text{gray}}(\mathbf{x}) = \frac{1}{3} \sum_{c \in R,G,B} \mathbf{x}_c, \quad f_{\text{color}}(\mathbf{x}) = \mathbf{x} - f_{\text{gray}}(\mathbf{x}). \tag{1}$$

Combining these generated samples with the color gallery from Geng et al. (2024a), we curated a total of 295 color illusion images.

**Multi-view.** Multi-view images produce different interpretations when observed from varying angles or orientations, illustrating viewpoint-dependent perceptual ambiguity. To generate such images with a diffusion model, we use two prompts: one for the original view ($0°$) and another for an alternate view ($90°$, $180°$, or $270°$). Combined with images collected online and those reported in Geng et al. (2024b), AmbiBench includes 249 multi-view images.

**Geometric.** Geometric images are illusions constructed from geometric patterns, such as impossible shapes or distorted proportions, that challenge spatial reasoning. They often exploit perspective

tricks, angular distortions, or size contrasts to create conflicting interpretations of spatial relationships. Classic examples include the Müller-Lyer illusion (Muller-Lyer, 1889). We curated 112 geometric ambiguous images from online sources and prior literature (Guan et al., 2024).

**Motion.** Motion illusions can be grouped into two categories. The first involves static patterns that evoke a strong impression of movement, typically through repetitive textures, color contrasts, or spatial arrangements, as in the rotating snakes illusion (Otero-Millan et al., 2012). The second involves dynamic stimuli that permit multiple interpretations of motion, producing perceptual ambiguity (Brooks & Barron, 2019). For this study, we collected motion-ambiguous stimuli from online sources and prior literature (Geng et al., 2024a), yielding 44 images and 18 videos.

### 3.2.2 SCENE-LEVEL CATEGORY

Multi-scene ambiguous images depict entire scenes that can be interpreted in multiple, often mutually exclusive ways. Unlike object-level illusions, which are ambiguous at the level of a single object, these images operate at the global scene level, where the same arrangement of elements can simultaneously support distinct scene interpretations. For example, as shown in Fig. 1, a composition may be perceived either as a portrait of a human face or as a collection of smaller figures (e.g., a farmer and a woman holding a child) arranged within a broader environment. A scene containing hidden content (objects) can also be regarded as a special case of a multi-scene image. Such examples highlight the competition between local object details and global scene organization in visual perception. We curated these images from online sources, resulting in a total of 38 multi-scene ambiguous samples.

### 3.2.3 MIXED CATEGORY

To increase the diversity and complexity of ambiguous perception, we introduce a new mixed category that combines features of both bistable and multi-scene ambiguity. Specifically, we take a bistable image as the reference and pair it with a scene-level prompt to generate a bistable-hidden multi-scene ambiguous image (i.e., a mixed image) using PTDiffusion (Gao et al., 2025b).

PTDiffusion builds on the off-the-shelf Latent Diffusion Model (LDM) (Rombach et al., 2022) and incorporates three main components: an inversion trajectory, a reconstruction trajectory, and a sampling trajectory. The inversion trajectory maps the reference image into the Gaussian noise space of the LDM. The reconstruction trajectory then restores the reference image from this inverted noise embedding. Finally, the sampling trajectory generates the illusion image from random noise under the guidance of the text prompt.

Images in this category exhibit layered ambiguity, where local details can alternate between multiple object-level interpretations (e.g., a duck or a rabbit in Fig. 1), while the global configuration simultaneously supports distinct scene-level perceptions (e.g., a rock cave). These mixed cases pose a greater challenge for visual inference, as observers must reconcile competing interpretations across both local and global scales while also managing cognitive ambiguity. After human filtering, AmbiBench includes 40 mixed ambiguous images.

### 3.2.4 TWO-STAGE FILTERING AND ANNOTATION

To ensure the quality and validity of ambiguous images, both collected and AI-generated images underwent a two-stage filtering and annotation process. In the first stage, a human annotator removed low-quality or trivial cases where the ambiguity was weak or not visually meaningful. For images collected from online sources, the annotator also assigned each sample to an appropriate ambiguity subcategory. In the second stage, three independent annotators reviewed the remaining images to confirm correct alignment with the designated ambiguity type and to verify that they were free of violent, harmful, or unsafe content. An image was discarded if more than one annotator judged it invalid. All human annotation was conducted in accordance with established ethical guidelines.

Through this rigorous process, we curated a high-quality, diverse collection of 2,250 ambiguous images spanning nine categories, which together form the foundation of AmbiBench.

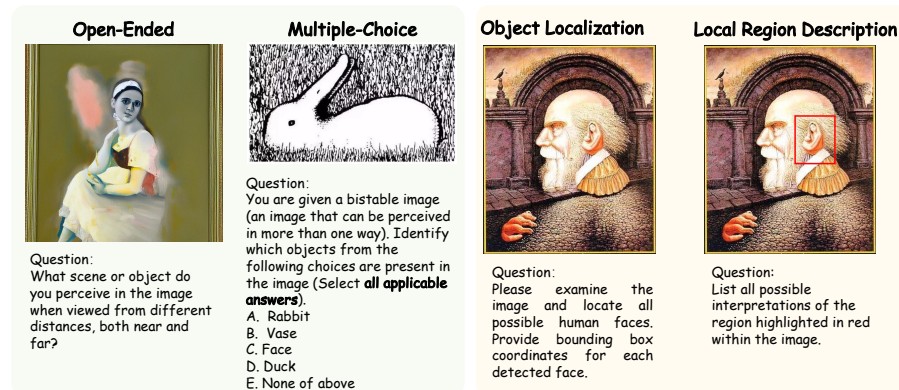

Figure 3: Examples of four task types on ambiguous images.

## 3.3 TASK DESIGN

To comprehensively evaluate the perceptual and cognitive abilities of VLMs on ambiguous images, we design a suite of tasks that probe different levels of understanding (examples shown in Fig. 3). Each ambiguity category may be paired with one or more task types.

### 3.3.1 AMBIGUOUS VISUAL UNDERSTANDING

**Open-ended.** Given an ambiguous image, the model is asked to generate a free-form response that reflects its interpretation. For instance, it may answer "What hidden objects can you see in the image?" for camouflaged cases, or "Is the gray line in the image horizontal?" for geometric illusions. This task evaluates the model's visual perception and reasoning, specifically its ability to recognize and respond to visual ambiguity.

**Multiple-choice.** In this task, the model must identify all valid interpretations from a set of candidates. It measures the ability to represent and maintain multiple percepts simultaneously, potentially aided by prior knowledge or contextual hints. For example, in bistable images, five choices are provided: two correspond to correct interpretations, two are plausible but incorrect distractors automatically generated by GPT-5, and one is "None." This setup challenges models to discriminate genuine perceptual alternatives from visually or semantically similar distractors.

### 3.3.2 FINE-GRAINED LOCAL-TO-GLOBAL AMBIGUOUS VISUAL REASONING

**Object Localization.** For images containing locally ambiguous objects that must be interpreted within a global context, we introduce an ambiguous localization task. The model is asked to identify ambiguous regions or objects with bounding boxes, such as all possible human faces in multi-scene images. This task evaluates fine-grained visual localization while also testing the model's ability to resolve both local and global ambiguities through spatial reasoning and imaginative inference.

**Local Region Description.** In multi-scene images, certain local elements can support multiple interpretations, for example, a human ear may also be perceived as a woman holding a baby in Fig. 3. By highlighting specific regions of interest, the model is required to generate all plausible interpretations. This task evaluates whether a VLM can integrate local and global cues in a manner consistent with human visual perception. Moreover, focusing on region-specific descriptions helps mitigate potential confounds, including the possibility that the full image appeared in pretraining.

## 4 EXPERIMENTS

In this section, we evaluate a broad set of large VLMs on AmbiBench, including both open-source and proprietary models. We begin by introducing the evaluation models together with a human baseline. We then analyze model performance across tasks and compare the results against human

responses. Finally, we investigate the internal mechanisms of VLMs in processing ambiguous perception, focusing on whether specialized attention heads are engaged during perceptual switching in bistable images, analogous to patterns observed in human cognition.

## 4.1 EXPERIMENTAL SETUP

**Models Evaluated.** We evaluate 12 state-of-the-art VLMs in a zero-shot setting, consisting of seven open-source models—Qwen2.5-VL-7B-Instruct, Qwen2.5-VL-72B-Instruct (Bai et al., 2025), InternVL3-8B-Instruct, InternVL3-78B-Instruct (Zhu et al., 2025), LLaVA-1.5-13B (Liu et al., 2023), Idefics3-8B-Llama3 (Laurençon et al., 2024), and Ovis2-34B (Lu et al., 2024b)—and five proprietary models: Gemini 2.5 Pro (Google DeepMind, 2025), GPT-5 (OpenAI, 2025), Kimi-VL-A3B-Thinking-2506 (Team et al., 2025b), Phi-4 Multimodal Instruct (Abdin et al., 2024), and Gemma-3-27B-IT (Team et al., 2025a). All models are evaluated on the full AmbiBench benchmark.

**Human Evaluation.** To enable a comprehensive comparison between VLMs and human perception on ambiguous images, we randomly sample 30 examples for hybrid and multi-view that contain more AI generated images, 10 examples for other categories for human evaluation. For the seven object-level categories, each image is paired with one open-ended question, while for the multi-scene and mixed categories, each image is paired with two questions addressing ambiguous localization and local region description. Each question is answered by at least three independent crowdworkers to ensure statistical reliability (more detail in Appendix A.4). All human evaluations were conducted in accordance with established ethical guidelines and received prior ethics approval.

For all object-level categories, we use open-ended questions to assess how models perceive ambiguity in images or videos. For the bistable category, we additionally include a multiple-choice task with prior information provided in the options. For the multi-scene and mixed categories, evaluation consists of open-ended questions, ambiguous localization, and local region description tasks, enabling a more comprehensive assessment. As a baseline, we also evaluate VLMs on non-ambiguous images (1,250 samples from the MS-COCO dataset (Lin et al., 2014; Le et al., 2019)).

For open-ended, multiple-choice, and local region description tasks, we employ an LLM judge (Qwen3-30B (Yang et al., 2025)) to verify alignment with ground truth, which is feasible given the simplicity of our labels (details in Appendix A.5). A response is considered correct only if all expected interpretations are provided (e.g., "near for young lady" and "far for rabbit" for the open-ended example in Fig. 3). For ambiguous localization, we adopt Average Precision (AP@0.5), treating a prediction as correct if the bounding box overlaps more than 50% with the ground truth. Examples of VLM and human test are shown in Appendix A.2 and A.4.

## 4.2 MAIN RESULTS

We present a comprehensive evaluation on AmbiBench. Detailed performance of SOTA VLMs and human participants across different ambiguity categories is reported in Table 1 and Table 2.

Table 1: Performance of 12 VLMs and humans on non-ambiguous (Non-Ambi.) cases and seven object-level ambiguous categories. For the multiple-choice (multi-label) task with five candidates and exactly two correct answers, the chance level is 10%.

| Model | Non-Ambi. | Camo. | Bistable | | Hybrid | Color | Multi-view | Geometric | Motion | | AVG |
|---|---|---|---|---|---|---|---|---|---|---|---|
| | | | open | choice | | | | | image | video | |
| Gemini 2.5 Pro | 82.48 | 84.04 | 25.45 | 85.45 | 15.94 | 46.78 | 18.40 | 69.89 | 76.00 | 27.78 | 53.22 |
| GPT5 | 92.80 | 86.05 | 29.09 | 81.82 | 18.12 | 53.22 | 18.00 | 66.54 | 48.00 | - | 54.84 |
| Qwen2.5-VL-7B-Instruct | 76.72 | 68.07 | 30.91 | 43.64 | 13.04 | 55.59 | 12.40 | 64.31 | 0.00 | 16.67 | 38.14 |
| Qwen2.5-VL-72B-Instruct | 90.32 | 79.12 | 34.55 | 56.36 | 15.94 | 45.08 | 13.60 | 64.68 | 36.00 | 11.11 | 44.68 |
| InternVL3-8B-Instruct | 85.60 | 63.68 | 18.18 | 25.45 | 12.32 | 52.50 | 12.80 | 56.51 | 8.00 | 22.22 | 35.73 |
| InternVL3-78B-Instruct | 89.52 | 83.07 | 36.36 | 67.27 | 18.84 | 62.71 | 21.60 | 62.83 | 44.00 | 72.22 | 55.84 |
| llava-hf/llava-1.5-13b-hf | 90.08 | 72.19 | 23.64 | 23.64 | 11.96 | 48.47 | 14.80 | 49.07 | 96.00 | 33.33 | 46.32 |
| Kimi-VL-A3B-Thinking-2506 | 84.88 | 73.68 | 14.55 | 50.91 | 15.94 | 54.92 | 12.00 | 66.54 | 40.00 | 38.89 | 45.23 |
| Phi-4-multimodal-instruct | 94.88 | 74.12 | 21.82 | - | 11.96 | 53.22 | 13.20 | 52.79 | 72.00 | - | 49.21 |
| Idefics3-8B-Llama3 | 72.48 | 72.89 | 25.45 | 9.09 | 8.73 | 55.25 | 14.40 | 49.44 | 8.00 | - | 35.08 |
| Gemma-3-27b-it | 89.76 | 76.49 | 40.00 | 54.55 | 14.13 | 47.80 | 20.40 | 51.67 | 76.00 | - | 52.31 |
| Ovis2-34B | 89.20 | 82.46 | 38.18 | 60.00 | 15.58 | 49.83 | 20.80 | 60.97 | 60.00 | 22.22 | 49.92 |
| Human | - | 86.67 | 100.00 | 80.00 | 63.33 | 76.67 | 51.72 | 83.33 | 76.67 | 86.67 | 74.45 |

**VLMs Performance.** As shown in Table 1 and Table 2, all evaluated VLMs perform substantially worse on ambiguous images than on natural, unambiguous ones, regardless of the ambiguity type. Humans maintain consistently high accuracy across categories, whereas VLMs fail to reach even half of human performance on five key categories: bistable, hybrid, multi-view, multi-scene, and mixed. For instance, while humans achieve 100% accuracy on bistable images, the best-performing VLM reaches only 40%, with Gemini 2.5 Pro and GPT-5 scoring 25.45% and 29.09%, respectively. The gap is even more pronounced in the mixed category: most VLMs fail completely, achieving 0% accuracy, while GPT-5—the strongest model—manages only 7.14% in detecting hidden bistable elements within mixed images. These results underscore both the difficulty of the task and the significant challenge posed by AmbiBench.

We further evaluate VLM performance on the dual interpretations of hybrid, multi-view, and color images. The analysis reveals that VLMs tend to prioritize high-frequency information (e.g., higher accuracy on near vs. far interpretations in hybrid images; Gemini 2.5 Pro: 80.80% on near vs. 17.39% on far), original orientations without rotation (e.g., higher accuracy at $0°$ than at rotated views; GPT-5: 68.80% without rotation vs. 18.80% with rotation), and colorful features (e.g., consistently higher accuracy on color than grayscale images across most models). Further details are provided in Appendix A.6.

**Performance Across Different Categories and Tasks.** While VLMs perform even worse on scene-level and mixed categories than on object-level ones—reflecting the greater perceptual ambiguity—we also observe substantial variability within the object-level tasks themselves. Specifically, models achieve relatively higher accuracy on camouflaged, color, and geometric categories, but their performance drops sharply on bistable, hybrid, and multi-view images. This discrepancy likely stems from differences in perceptual demands: camouflaged images can often be addressed through feature-level separation (e.g., distinguishing objects from background clutter), while the color category is simplified by our design, which provides paired subfigures in both color and grayscale. In contrast, bistable, hybrid, and multi-view illusions require reconciling conflicting global configurations, maintaining multiple plausible interpretations within a single image, or performing perceptual adjustments such as zooming, remain highly challenging for current VLMs.

Further analysis reveals that even for three variants of the classic rabbit–duck illusion in different painting styles, some VLMs can correctly identify only one or two, with outputs varying according to low-level image features. This suggests that VLMs rely primarily on feature-level processing rather than the global integration characteristic of mature human perception (Taranu et al., 2019). Notably, camouflaged, geometric, and bistable images are mostly drawn from existing datasets, and prior exposure may partly explain the stronger performance on camouflaged and geometric illusions. For bistable cases, we designed prompts requiring two distinct interpretations, ensuring that models must demonstrate multi-percept reasoning rather than recalling a single memorized answer.

We observe clear performance discrepancies across tasks. On bistable images, most VLMs perform better in the multiple-choice setting than in open-ended responses, as the correct answers are explicitly presented as prior information. In contrast, on multi-scene images, VLMs perform worse on ambiguous localization and local region description than on open-ended questions. This highlights the limitations of current VLMs in global processing, while their relatively stronger open-ended performance may partly reflect prior exposure to similar images during pretraining. Moreover, most models achieve higher accuracy on localization tasks involving local single objects than combined ones (e.g., "woman" vs. " old man face" in ambiguous localization in Fig. 3), reinforcing that ambiguous perception remains a core challenge. The consistently low performance on localization tasks further underscores the difficulty of detecting and reasoning about objects in layered scenes.

**Perception Catering to Human Preference.** Interestingly, VLMs achieve high performance on motion illusions when asked whether they perceive motion in static images. Although these images contain no actual movement, VLMs frequently report perceiving motion, thereby aligning with human illusory perception rather than physical reality. This suggests that VLMs may have implicitly learned perceptual priors from training data that mirror human biases, leading them to "see" what humans see—even when it is an illusion. The comparison with the MS-COCO static-image setting (Lin et al., 2014; Le et al., 2019) and the static prompts in Appendix A.7 further supports this. Such behavior highlights the human-centric nature of large VLM perception, while also reminding us that human perception itself is not always faithful to the physical world (Zhang et al., 2023).

Table 2: Performance of 12 VLM models and humans across multi-scene and mixed categories.

| Model | Multi-scene | | | | Mixed | | | AVG |
|---|---|---|---|---|---|---|---|---|
| | Open | loc(single) | loc(combined) | Region | Open | localization | Region | |
| Gemini2.5pro | 60.53 | 10.53 | 15.09 | 35.00 | 5.00 | 30.00 | 7.50 | 23.52 |
| GPT5 | 78.95 | 36.84 | 26.43 | 27.50 | 7.50 | 10.00 | 10.00 | 28.28 |
| Qwen2.5-VL-7B-Instruct | 23.68 | 47.37 | 12.83 | 27.50 | 0.00 | 10.00 | 0.00 | 19.89 |
| Qwen2.5-VL-72B-Instruct | 42.11 | 55.26 | 17.74 | 25.00 | 0.00 | 0.00 | 0.00 | 20.87 |
| InternVL3-8B-Instruct | 31.58 | 0.00 | 0.00 | 37.50 | 0.00 | 0.00 | 0.00 | 9.87 |
| InternVL3-78B-Instruct | 34.12 | 21.05 | 5.79 | 25.00 | 2.50 | 0.00 | 0.00 | 12.71 |
| llava-hf/llava-1.5-13b-hf | 28.95 | 0.00 | 0.00 | 32.50 | 0.00 | 0.00 | 2.50 | 9.28 |
| Kimi-VL-A3B-Thinking-2506 | 36.84 | 7.89 | 6.14 | 20.00 | 0.00 | 0.00 | 0.00 | 10.41 |
| Phi-4-multimodal-instruct | 21.05 | 0.00 | 0.00 | 30.00 | 0.00 | 0.00 | 0.00 | 7.15 |
| Idefics3-8B-Llama3 | 28.95 | 7.89 | 12.28 | 35.00 | 0.00 | 10.00 | 2.50 | 14.23 |
| Gemma-3-27b-it | 47.34 | 23.68 | 11.40 | 40.00 | 0.00 | 7.50 | 7.50 | 19.78 |
| Ovis2-34B | 18.42 | 23.68 | 3.88 | 31.25 | 0.00 | 0.00 | 0.00 | 11.26 |
| Human | - | 56.67 | 43.33 | 80.00 | - | 76.67 | 60.00 | 63.33 |

**AmbiBench VS Other Benchmark.** We compare AmbiBench with other benchmarks, including IllusionVQA Shahgir et al. (2024) and HallusionBench Guan et al. (2024), which also contain color ambiguity. Results show that most VLMs struggle with this type of ambiguity, while AmbiBench poses an even greater challenge (see Table 13 in Appendix A.6).

Further statistical analyses for robustness are provided in Appendix A.6, and additional visualization analyses including bistable and multi-scene examples are presented in Appendix A.8.

### 4.3 PERCEPTUAL-SWITCH HEADS OF VLMS

Bistable perception is a compelling example of cognitive self-organization, offering a model for studying how "the brain makes up its mind". Research on human bistable perception (Knapen et al., 2011) shows that frontal and parietal regions in the right hemisphere are specifically engaged during perceptual switches. Inspired by this, we examine whether VLMs display distinctive activation patterns, when transitioning between alternative interpretations of the same ambiguous input.

We evaluate four VLMs, including InternVL3 and Qwen2.5-VL, across multiple model scales. Each model is first prompted to provide a single interpretation of a bistable or multi-scene image, and the corresponding attention head activations are recorded as $X = \{x_l^m \mid l = 1, \ldots, N_l, \ m = 1, \ldots, N_h\}$. The model is then prompted to generate an alternative interpretation by providing a hint that the input is bistable. If the model successfully switches interpretations, we record the updated head activations as $\bar{X} = \{\bar{x}_l^m \mid l = 1, \ldots, N_l, \ m = 1, \ldots, N_h\}$. To quantify the role of each head in perceiving ambiguity (i.e., representing per-

Table 3: Negative Intervention of different VLMs on Bistable and Multi-Scene stimuli, the score is LLM-Judge Accuracy.

| Models | Inter_Head | Bistable | Multi-Scene |
|---|---|---|---|
| Qwen2.5-VL-3B-Instruct | random | 38.18 | 55.26 |
| | cognitive | 0.00 | 2.63 |
| Qwen2.5-VL-7B-Instruct | random | 41.82 | 55.26 |
| | cognitive | 0.00 | 0.00 |
| InternVL3-2B | random | 29.09 | 63.16 |
| | cognitive | 0.00 | 0.00 |
| InternVL3-8B | random | 38.18 | 63.16 |
| | cognitive | 0.00 | 0.00 |

ceptual switching (Knapen et al., 2011; Leopold & Logothetis, 1999)), we compute a *switching score* defined as the cosine similarity between pre- and post-switch activations across all bistable images. Heads with low similarity are designated as *perceptual-switch heads*.

Figure 4 shows the heatmap of attention-head switching for bistable images, revealing several consistent properties of perceptual-switch heads. Across all examined models, a subset of heads is reliably associated with ambiguity switching, yet the process remains strikingly sparse. For instance, in InternVL3-8B, only about ten heads (1%) achieve a switching score above 0.8. These heads emerge intrinsically, independent of model architecture or scale, and are predominantly concentrated in middle layers, underscoring their central role in encoding perceptual switching. Results on multi-scene images can be found in Fig. 16. The high Pearson correlations between head-activation heatmaps across bistable and multi-scene categories further indicate that perceptual-switch heads are intrinsic, independent of data category (see Appendix A.9 for details). Larger models consistently have more perceptual-switch heads, reflecting their superior performance in Tables 1 and 2.



Figure 4: Perceptual-switch heads in VLMs mediate switching between two interpretations for bistable stimuli.

## 4.4 INFLUENCE OF PERCEPTUAL-SWITCH HEADS

In this section, we investigate how perceptual-switch heads influence the perception of VLMs on bistable and multi-scene through both negative interventions (masking out perceptual-switch heads) and positive interventions (shifting heads toward perceptual-switch function).

**Negative Intervention:** After identifying the perceptual-switch heads associated with each function, we examine their functional role by evaluating the model's behavior on bistable and multi-scene ambiguous images under targeted interventions. We perform head ablation by scaling the output of a specific attention head with a small factor $\epsilon$ (e.g., 0.001), effectively suppressing its contribution:

$$x_i^{\text{mask}} = \text{Softmax}\left(\frac{W_q^i W_k^{iT}}{\sqrt{d_k/n}}\right) \cdot \epsilon W_v^i \quad (2)$$

Table 3 shows that masking perceptual-switch heads leads to a significant performance drop across all models, while masking the same amount of random head lead to margin performance loss.

Table 4: Positive Intervention of different VLMs on Bistable and Multi-Scene stimuli, the score is LLM-Judge Accuracy.

| Models | Inter_Head | Bistable | Multi-Scene |
|---|---|---|---|
| Qwen2.5-VL-3B-Instruct | Before | 43.64 | 57.89 |
| | After | **45.45** | **60.53** |
| Qwen2.5-VL-7B-Instruct | Before | 41.82 | 63.16 |
| | After | **45.45** | **68.42** |
| InternVL3-2B | Before | 29.09 | 63.16 |
| | After | **41.82** | **65.79** |
| InternVL3-8B | Before | 36.36 | 60.53 |
| | After | **38.18** | 60.53 |

**Positive Intervention:** We calculate the activation directions of identified perceptual-switch head as: $\text{dir}_l^h = \mathbb{E}_{i \in \mathcal{D}_{\text{correct}}}\left[x_l^h(i)\right] - \mathbb{E}_{i \in \mathcal{D}_{\text{incorrect}}}\left[x_l^h(i)\right]$, where $x_l^h(i)$ denotes the activation of head at layer $l$ and index $h$, and $\mathcal{D}_{\text{correct}}$ and $\mathcal{D}_{\text{incorrect}}$ represent the sets of samples answered correctly and incorrectly, respectively. Then we estimate the standard deviation of activations (Li et al., 2023) along the cognitive function direction to be $\sigma_l^h$, and shift original head activation as $x_l^h(i) \leftarrow x_l^h(i) + \alpha \sigma_l^h \text{dir}_l^h$, where $\alpha$ is a parameter. We tune $\alpha \in \{0.1, 0.3, 0.5\}$.

The experimental results in Table 4 show that enhancing the activation of functional heads along their corresponding functional directions improves performance. For example, positive intervention on perceptual-switch heads in InternVL3-2B increased accuracy on bistable from 29.09% to 41.82%.

## 5 CONCLUSION

We introduce AmbiBench, a benchmark for evaluating VLMs on ambiguous images. Our large-scale study of 12 state-of-the-art VLMs shows that, despite strong performance on conventional benchmarks, current models struggle with perceptual ambiguity: they achieve low accuracy, exhibit limited interpretive diversity, and tend to default to contextually dominant labels. These findings expose a gap between machine perception and human cognition, particularly in abstract interpretation and considering multiple perspectives, positioning AmbiBench as a foundation for developing more cognitively aligned, ambiguity-aware VLMs. Furthermore, improvements on AmbiBench relate to broader visual abilities such as spatial intelligence (e.g., hybrid images with spatial ambiguity) and compositional reasoning (e.g., multi-scene images requiring integration of local and global cue). Although AmbiBench is designed purely for evaluation, uneven category distributions may cause certain model weaknesses to appear more or less pronounced depending on sample count.

ETHICS STATEMENT

This work evaluates large vision-language models on ambiguous images. AmbiBench contains only publicly available or research-curated ambiguous stimuli and does not include personal or sensitive data. The evaluation process was designed to be transparent and reproducible, following established standards of research integrity and ethical conduct. All human annotations and user studies were conducted under institutional ethics approval, and no personally identifiable information was collected, used, or processed in this study.

REPRODUCIBILITY STATEMENT

We provide images included in AmbiBench at https://anonymous.4open.science/r/ambibench-25A6/. Upon publication, we will release the full dataset and evaluation scripts to facilitate further research and ensure full reproducibility of our results.

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

# A APPENDIX

## A.1 AMBIBENCH CATEGORIES

Table 5 summarizes the nine categories in AmbiBench, covering object-level, scene-level, and mixed types, along with detailed descriptions. Table 6 presents a comparison of AmbiBench with other datasets containing ambiguous images.

Table 5: Categories of ambiguous visual data in **AmbiBench**, organized by major type.

| Major Type | Category | Description |
|---|---|---|
| **Object-Level** | Camouflaged | Natural images containing objects deliberately hidden within complex backgrounds, making them difficult to detect. |
| | Bistable | Images allowing two mutually exclusive interpretations, where perception alternates between different stable states. |
| | Hybrid | Images blending real and artistic components, often embedding two distinct scenes or objects at different spatial frequencies. |
| | Color | Optical illusions or artworks whose interpretation changes when viewed in color versus grayscale. |
| | Multi-view | Images yielding different interpretations when viewed from varying angles or orientations. |
| | Geometric | Visual illusions based on geometric patterns (e.g., impossible shapes, distorted proportions) that mislead perception. |
| | Motion | Static or dynamic images that evoke illusory motion or conflicting motion cues. |
| **Scene-Level** | Multi-scene | Scene-level ambiguity where a single image contains multiple possible scene interpretations. |
| **Mixed** | Mixed ambiguous | Newly introduced category where two types of ambiguity coexist in a single image; i.e., bistable and multi-scene. |

Table 6: Comparison of AmbiBench with existing benchmarks on ambiguous images.

| Dataset | #Images | Ambiguity Types | Modalities | Task Types | Human Study |
|---|---|---|---|---|---|
| HallusionBench (Guan et al., 2024) | 72 | color, geometric | Image, video | multiple-choice | ✓ |
| Grounding-Illusions (Zhang et al., 2023) | ∼100 | color, geometric | image | open-ended, multiple-choice, localizaton | ✗ |
| SemVINK (Li et al., 2025) | 112 | multi-scene (single hidden) | image | multiple-choice, open-ended | ✗ |
| IllusionVQA (Shahgir et al., 2024) | ∼400 | color, multi-scene(hidden), geometric | image | multiple-choice, open-ended, soft-Localization | ✓ |
| IllusionBench+ (Zhang et al., 2025) | 1051 | geometric, bistable | images | multiple-choice, open-ended | ✓ |
| **AmbiBench (Ours)** | **> 2000** | **9 Types (incl. Mixed)** | **Image, Video** | **multiple-choice, open-ended, localization, region-based description** | **✓** |

## A.2 EXAMPLE PROMPTS OF VLM TEST FOR DIFFERENT CATEGORIES ON FOUR TASKS

Table 7: Example prompts of ambiguous visual data in **AmbiBench**, grouped by major type.

| Category | Question Type | Prompt |
|---|---|---|
| **Camouflaged** | Camo | You are a camouflage detection expert. Carefully examine the image and locate the camouflaged object, provide the object name in one word. Output ONLY in the following format: Object: 'object name in one word'. |
| | Noncamo | You are a camouflage detection expert. Look at the image carefully and identify what one main object in the scene. The output is ONLY in the following format: Object: 'object name in one word'. |
| **Bistable** | Open | You are given a bistable image (an image that can be perceived in more than one way). Identify two possible distinct objects or visual features that a person might recognize in this image. Respond with only the object words, separated by commas, like 'Object1, Object2', without explanations or extra words. |
| | Choice | You are given a bistable image (an image that can be perceived in more than one way). Identify which objects from the following choices are present in the image. Select **all applicable answers** by returning their **letters only**, separated by commas (e.g., A, C). If none of the objects are present, choose **E**. Choices: A: car with snow, B: face, C: Motorcycle, D: Bus, E: None Your answer: |
| **Hybrid** | | This is a hybrid image. What two different objects or scenes can be perceived in it, one when focusing on fine details (Object 1) and the other when viewed from a distance (Object 2)? Respond with only two object words, separated by commas, in the form of 'Object1, Object2', where Object1 is the one focusing on fine details, and Object2 is the one viewing from a distance. |
| **Color** | | This image contains two subimages with different colors. The left and right subimages depict different objects or scenes. Respond with exactly two words or two phrases, separated by a comma, in the format 'Object1, Object2', where Object1 describes the left subimage and Object2 describes the right subimage. |
| **Multi-view** | | You are given an image. Look at the image normally and describe what object or scene you see. Then imagine rotating the image (90 degrees or 180 degrees, whichever makes sense) and describe what new object or scene it looks like after rotation. Respond with exactly two words or two short phrases, separated by a comma, in the format 'Object1, Object2'. Do not add extra words or explanations. |
| **Geometric** | | Is the gray line in the image horizontal? The answer should be short and concise, and without explanations. |

**Table 7 – continued from previous page**

| Category | Question Type | Prompt |
|---|---|---|
| **Motion** | Image | Do you think the scene in the image is moving? Respond ONLY with Yes or No. |
| | Video | You are given an video. Watch the video and describe what object or scene you see. Respond with exactly one word or one short phrase. Do not add extra words or explanations. |
| **Multi-scene** | Open | You are given a multi-layer optical illusion image that contains two layers of meaning, where each layer represents a different interpretation of the same image. List all possible objects or scenes for both layers. Use short words or phrases and separate two layers with a semicolon ';'. Format your answer strictly as: 'layer1 objects; layer2 objects' Do not add any extra explanation. |
| | Localization | You are a object detection expert. Carefully examine the image and locate the position of all the possible objects in the image, provide the bounding box coordinates of all the elephants, the image size is Width: 802, Height: 470. If only one exists in the image, output in the following format: '$[(x\_min, y\_min, x\_max, y\_max)]$', if multiple exist in the image, output in the following format: '$[(x\_min, y\_min, x\_max, y\_max),$ $(x\_min, y\_min, x\_max, y\_max), ...]$', if no elephant exist in the image, return 'None'. |
| | Region | You are given a multi-layer optical illusion image that contains two layers of meaning in the green bounding box, where each layer represents a different interpretation of the same image. Within the **green bounding box** in the image, list all possible objects or scenes for both layers do not interpret objects outside the bounding box. Use short words or phrases and separate two layers with a semicolon ';'. Format your answer strictly as: 'layer1 objects; layer2 objects' Do not add any extra explanation. |
| **Mixed** | Open | You are given an image that contains one or more 'hidden objects', these objects can only be recognized by carefully observing their outlines, edges, or subtle shapes. By looking at the image carefully, list all possible objects or scenes that are hidden in the image. If you can't find any hidden objects, just list all possible objects or scenes that existing in the image. Use short words or phrases, separate with commas, format your answer strictly as: 'object1, object2', do not add any extra explanation. |
| | Localization | You are a object detection expert. You are given an image with some objects existing in the image. Carefully examine the image and locate the position of the young woman, old woman in the image by providing the bounding box coordinates, the image size is Width: image width, Height: image height. Output exactly in the following format: '$[(x\_min, y\_min, x\_max, y\_max)]$', do not add any extra explanation. If you can't find young woman, old woman in the image, return 'None'. |

**Table 7 – continued from previous page**

| Category | Question Type | Prompt |
|---|---|---|
| | Region | You are given an multi-layer optical illusion image that contains two layers of meaning in the green bounding box, where each layer represents a different interpretation of the same image. Within the \*\*green bounding box\*\* in the image, list all possible objects or scenes for both layers do not interpret objects outside green bounding box. Use short words or phrases and separate two layers with a semicolon ';'. Format your answer strictly as: 'layer1 objects; layer2 objects' Do not add any extra explanation. |

### A.3 OVERVIEW OF BASELINE MODELS

- **Gemini2.5pro**(Comanici et al., 2025) employs a unified multimodal transformer jointly trained on text, images, and code. It introduces long-context optimization and cross-modal attention mechanisms. These innovations enable strong reasoning across diverse inputs and extended sequences.

- **GPT5**(OpenAI, 2025) extends the GPT architecture with multimodal adapters and hierarchical attention. It leverages mixture-of-experts routing for efficiency and improved specialization. The model emphasizes structured reasoning and robust performance on complex multimodal tasks.

- **Qwen2.5-VL-7B-Instruct**(Bai et al., 2025) combines a vision encoder with a 7B-parameter LLM backbone. Instruction-tuned multimodal alignment ensures effective cross-modal understanding. Its lightweight design enables efficient inference under constrained resources.

- **Qwen2.5-VL-72B-Instruct**(Bai et al., 2025) scales Qwen to 72B parameters with deeper cross-attention layers. It couples large-scale pretraining with fine-grained instruction tuning. This balance yields strong performance in multilingual reasoning and visual understanding.

- **InternVL3-8B-Instruct**(Zhu et al., 2025) integrates a vision transformer with a multimodal fusion module on top of an 8B LLM. It adopts native multimodal pretraining rather than post-hoc alignment. The design emphasizes reproducibility and efficient large-scale training.

- **InternVL3-78B-Instruct**(Zhu et al., 2025) is a 78B-parameter version with hierarchical cross-modal fusion. It leverages multi-stage instruction tuning to improve compositional reasoning. This yields stronger grounding and long-horizon multimodal performance.

- **LLaVA-1.5-13B**(Liu et al., 2024) aligns a pretrained CLIP vision encoder with a 13B LLaMA backbone through a projection layer. It uses visual instruction tuning on high-quality synthetic datasets. This iterative approach achieves strong results in image-grounded dialogue and reasoning.

- **Kimi-VL-A3B-Thinking-2506**(Team et al., 2025b) employs a multimodal mixture-of-experts backbone with chain-of-thought prompting. The model is supervised with explicit reasoning traces. These innovations improve robustness and performance on multi-step visual–language tasks.

- **Phi-4-multimodal-instruct**(Abdin et al., 2024) is a compact 5.6B-parameter multimodal transformer. It relies on high-quality filtered and synthetic textbook-style data. This data-centric design enables cost-efficient scaling with strong generalization.

- **Idefics3-8B-LLaMA3**(Laurençon et al., 2024) integrates LLaMA3 with a SigLIP vision encoder using dual attention. Each image is represented with a compact set of tokens in a shared latent space. The model emphasizes modularity, long-context support, and open-source reproducibility.

- **Gemma-3-27B-it**(Team et al., 2025a) is a 27B decoder-only transformer with interleaved local–global attention. It incorporates a SigLIP vision encoder and Pan & Scan for flexible image inputs. Distillation and RLHF post-training enhance reasoning, math, and multilingual abilities.

- **Ovis2-34B**(Lu et al., 2024b) features a 34B-parameter multimodal transformer with a multi-branch fusion network. It emphasizes robust multimodal pretraining on large-scale, community-curated datasets. These design choices yield competitive performance on diverse vision–language tasks.

### A.4 QUESTIONNAIRE FOR HUMAN (EXAMPLES)

Figures 5,6,7,8,9,10,11 show the examples for human evaluation test for different tasks.

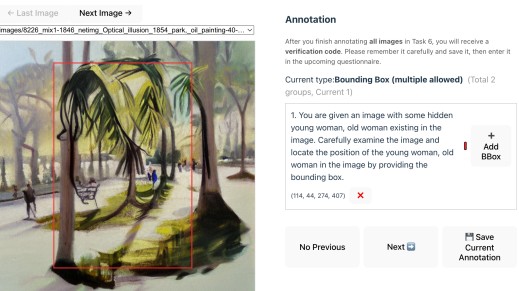

Figure 5: Human annotation website, case 1.

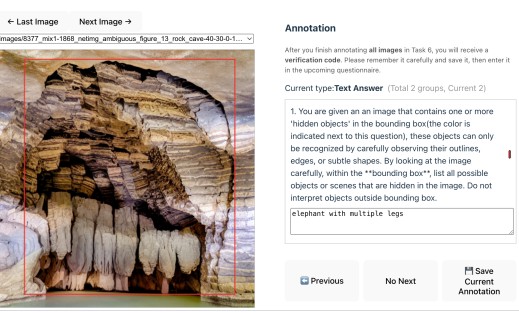

Figure 6: Human annotation website, case 2.

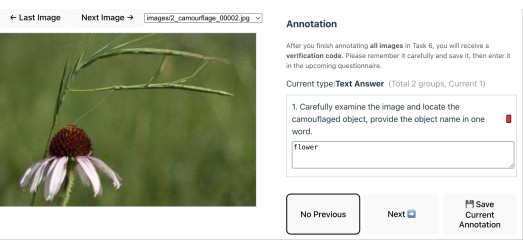

Figure 7: Human annotation website, case 3.

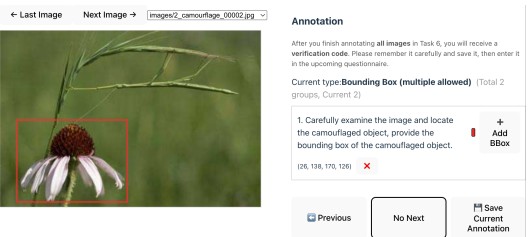

Figure 8: Human annotation website, case 4.

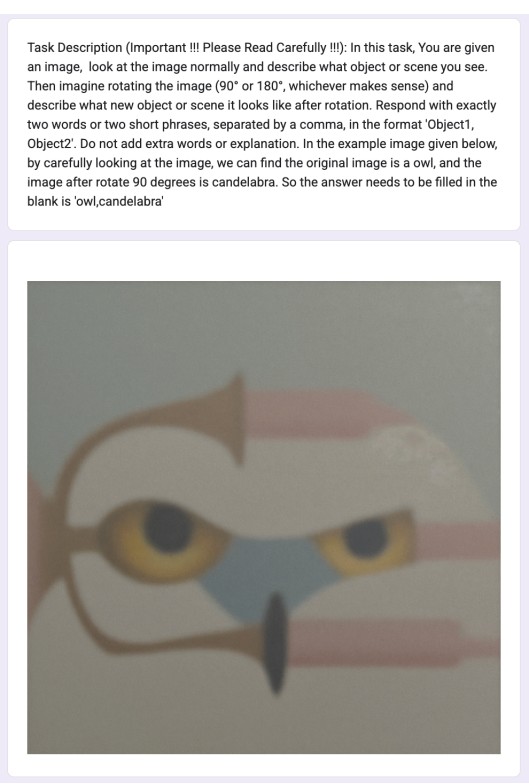

Figure 9: Human study demo.

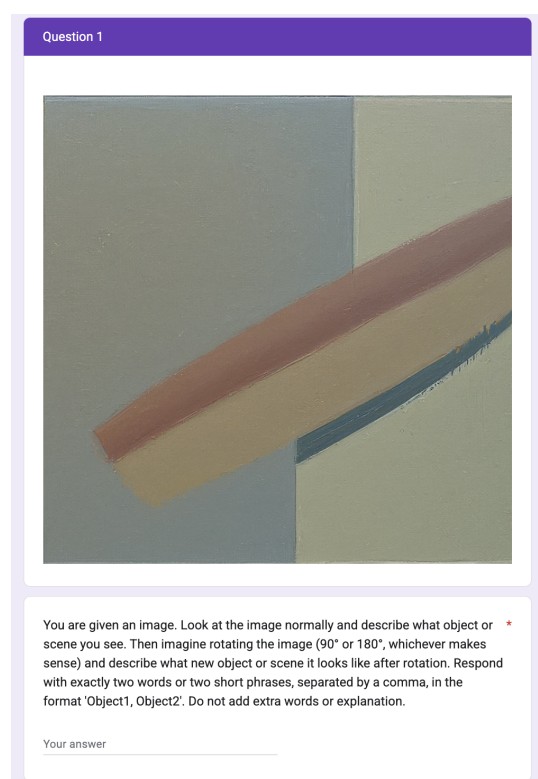

Figure 10: Human study question case 1.

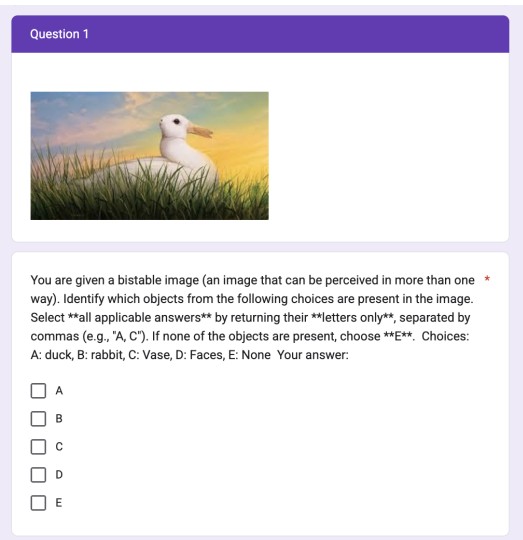

Figure 11: Human study question case 2.

**Further analysis of human performance on hybrid images.** Hybrid images containing abstract artwork are challenging for both VLMs and humans. We demonstrate the value of our generated hybrid images through the following analyses:

**1. Novelly generated stimuli show similar performance to stimuli from online sources for both humans.** For hybrid images in AmbiBench, 29 images were selected from online sources. We also randomly selected the same number of newly generated hybrid images. The same three participants evaluated both subsets of hybrid images. As shown in Table 8, the mean human accuracy across the

three participants was 67.82% for Hybrid (online source) and 63.22% for Hybrid (newly generated). This indicates that our novel stimuli generated using diffusion models are valid. The examples of hybrid images at the provided link are drawn from both online sources and newly generated images.

**2. Meaningful ambiguous images also differ across individuals.** In addition, we note that human performance exhibits substantial variability across participants. For example, among the three participants, one may achieve over 80% accuracy while another achieves around 60%. Ambiguous images that rely on abstract interpretation or imagination remain challenging for humans, and differences in background and prior knowledge among individuals contribute to this variability. A classic example is The Dress (2015), in which different observers perceived different colors under the same viewing conditions, illustrating that even well-controlled stimuli can elicit divergent perceptual experiences.

Table 1 reports human performance on 30 hybrid and multi-view images, which we find to be sufficient for reliable human evaluation.

Table 8: Human Performance on more hybrid and multi-view images.

| **Human** | Hybrid (online source) | Hybrid (new generated) | Multi-view |
|---|---|---|---|
| 1 | 82.75 | 58.62 | 68.97 |
| 2 | 58.62 | 65.62 | 41.38 |
| 3 | 62.06 | 65.62 | 44.83 |
| Average | 67.82 | 63.22 | 51.72 |

## A.5 DETAILS OF LLM-JUDGE

We provide the LLM judge prompt along with several illustrative examples.

---

**Prompt**

**LLM Judge Prompt:**
You are a linguistic expert. Compare the following two sets of words and determine whether **every word** in Set 1 can be matched with **exactly one word** in Set 2 such that they meet **any** of these relationships:
1. They are **synonyms** (same or very similar meaning), e.g., "car" and "automobile".
2. They have a **hypernym/hyponym** relationship (one is a broader or narrower category of the other), e.g., "fruit" and "apple".
3. They are **exactly the same word**.
Notes: - The order of words does not matter.
- Each word in Set 1 must have a unique matching word in Set 2.
- If **any word** in Set 1 cannot be matched, return: False.
- Only if **all words** can be matched, return: True.
Set 1: `predicted`
Set 2: `ground_truth`
Answer with only one word: True or False.

---

**Examples:**

Predicted: Insect; Ground Truth: Insect; Response: True

Predicted: Insect; Ground Truth: Spider; Response: True

Predicted: Insect; Ground Truth: Leaves; Response: False

## A.6 MORE ANALYSIS

**More statistical analysis for robustness.** To ensure the stability and reproducibility of our experimental results, we configure the VLMs in a deterministic mode (i.e., False for stochastic sampling), following standard benchmarking practices for fairness.

We additionally evaluate each model under its original sampling settings (temperature = 1) and repeat the experiments five times with different random seeds. The results are summarized in the following table. We evaluate performance across object-level categories (color, multi-view, and hybrid), as well as scene-level and mixed-level categories. The overall trends are consistent with those reported in Table 1 and Table 2 of the main paper.

A t-test comparing performance across these categories confirms that our findings—specifically, that VLMs perform worse on scene-level and mixed categories than on object-level categories by multiple testing(p-value$\ll$0.05).

Meanwhile, we compare GPT5 and Gemini 2.5 Pro with Qwen2.5-VL-7B-Instruct using multiple-testing correction across several categories. The results in Table 11 show that for four categories, both GPT5 and Gemini 2.5 Pro outperform Qwen2.5-VL-7B with statistical significance. However, for the more challenging mixed category, the performance differences between models are not statistically significant.

Table 9: Performance of 12 VLMs on different categories (mean and variant over 5 seeds).

| Model | Multi-Scene (open) | Mix (open) | Color | Multi-View | Hybrid |
|---|---|---|---|---|---|
| Gemini 2.5 Pro | $63.68 \pm 7.33$ | $1.43 \pm 1.75$ | $48.27 \pm 1.19$ | $19.76 \pm 1.23$ | $18.33 \pm 0.96$ |
| GPT5 | $74.74 \pm 2.11$ | $1.43 \pm 3.19$ | $51.59 \pm 0.92$ | $16.24 \pm 1.20$ | $20.58 \pm 1.12$ |
| Qwen2.5-VL-7B-Instruct | $26.32 \pm 0.00$ | $0.00 \pm 0.00$ | $47.53 \pm 4.55$ | $9.44 \pm 1.49$ | $9.15 \pm 2.31$ |
| Qwen2.5-VL-72B-Instruct | $37.89 \pm 2.11$ | $0.00 \pm 0.00$ | $41.49 \pm 2.31$ | $9.76 \pm 1.92$ | $13.19 \pm 1.74$ |
| InternVL3-8B-Instruct | $23.16 \pm 4.21$ | $0.00 \pm 0.00$ | $49.36 \pm 1.76$ | $9.20 \pm 1.81$ | $11.34 \pm 0.69$ |
| InternVL3-78B-Instruct | $38.95 \pm 1.05$ | $0.00 \pm 0.00$ | $57.97 \pm 2.39$ | $16.48 \pm 2.56$ | $14.53 \pm 2.15$ |
| llava-hf/llava-1.5-13b-hf | $21.05 \pm 4.08$ | $0.00 \pm 0.00$ | $30.89 \pm 8.92$ | $12.56 \pm 1.12$ | $10.08 \pm 1.12$ |
| Kimi-VL-A3B-Thinking-2506 | $32.63 \pm 2.11$ | $0.00 \pm 0.00$ | $48.63 \pm 1.59$ | $11.52 \pm 1.13$ | $14.83 \pm 0.72$ |
| Phi-4-multimodal-instruct | $12.63 \pm 4.21$ | $0.00 \pm 0.00$ | $46.92 \pm 3.66$ | $12.56 \pm 0.32$ | $10.36 \pm 0.64$ |
| Idefics3-8B-Llama3 | $27.37 \pm 1.29$ | $0.00 \pm 0.00$ | $48.47 \pm 3.39$ | $10.08 \pm 2.08$ | $7.90 \pm 0.41$ |
| Gemma-3-27b-it | $54.74 \pm 1.05$ | $0.00 \pm 0.00$ | $49.29 \pm 0.41$ | $19.44 \pm 0.48$ | $15.07 \pm 0.29$ |
| Ovis2-34B | $26.84 \pm 4.21$ | $0.00 \pm 0.00$ | $51.25 \pm 0.14$ | $17.28 \pm 1.76$ | $16.23 \pm 0.14$ |

Table 10: Bonferroni-corrected p-values (group size = 3 per set).

| Model | Color vs Multi-Scene | Multi-View vs Multi-Scene | Hybrid vs Multi-Scene | Color vs Mix | Multi-View vs Mix | Hybrid vs Mix |
|---|---|---|---|---|---|---|
| Gemini 2.5 Pro | 4.26e-06 | 6.39e-10 | 4.77e-08 | 3.18e-22 | 1.93e-11 | 3.81e-09 |
| GPT5 | 1.34e-11 | 1.83e-10 | 5.37e-10 | 2.14e-19 | 1.47e-11 | 6.39e-05 |
| Qwen2.5-VL-7B | 3.94e-01 | 2.38e-04 | 3.15e-03 | 7.56e-06 | 1.87e-06 | 1.43e-03 |
| Qwen2.5-VL-72B | 5.04e-02 | 1.53e-05 | 1.17e-03 | 5.94e-07 | 7.53e-10 | 1.20e-04 |
| InternVL3-8B | 7.95e-03 | 2.49e-06 | 9.81e-05 | 1.03e-08 | 1.10e-07 | 4.08e-06 |
| InternVL3-78B | 7.80e-07 | 8.76e-06 | 3.69e-05 | 5.73e-10 | 1.18e-07 | 1.35e-03 |
| LLaVA-1.5-13B | 2.05e-01 | 3.66e-02 | 9.87e-02 | 7.77e-04 | 3.66e-04 | 1.39e-05 |
| Kimi-VL | 7.74e-03 | 1.04e-07 | 6.36e-04 | 4.32e-15 | 1.24e-05 | 1.58e-07 |
| Phi-4-MM | 6.44e-01 | 1.34e-01 | 2.38e-02 | 2.91e-06 | 6.27e-06 | 6.87e-06 |
| Idefics3-8B | 2.88e-02 | 1.17e-07 | 9.87e-06 | 3.66e-13 | 3.72e-06 | 4.26e-06 |
| Gemma-3-27B | 2.28e-09 | 8.67e-12 | 3.30e-10 | 1.01e-15 | 6.15e-09 | 5.13e-08 |
| Ovis2-34B | 7.99e-03 | 1.22e-07 | 2.02e-04 | 8.43e-11 | 2.90e-07 | 2.59e-09 |

Table 11: Bonferroni-corrected p-values of comparing GPT5 / Gemini2.5-Pro against Qwen2.5-VL-72B-Instruct.

| Model | Multi-Scene | Mix | Color | Multi-View | Hybrid |
|---|---|---|---|---|---|
| GPT5 | 0.000 | 1.865 | 0.00106 | 0.002185 | 0.000525 |
| Gemini2.5-Pro | 0.004365 | 0.71 | 0.00565 | 0.000145 | 0.00515 |

Table 12 shows the preferences of VLMs for the two possible interpretations of hybrid, color, and multi-view images. We observe that these preferences align with those of humans. The questionnaire for human participants uses the same question description as those provided to the VLMs.

Table 12: Performance of 12 VLM models and humans across multi-scene and mixed categories.

| Model | Hybrid | | | Color | | | Multi-view | | |
|---|---|---|---|---|---|---|---|---|---|
| | Combined | Near | Far | Combined | Color | Gray | Combined | w/o Rotation | w Rotation |
| Gemini2.5pro | 15.94 | 80.80 | 17.39 | 46.78 | 74.58 | 63.39 | 18.40 | 66.80 | 18.80 |
| GPT5 | 18.12 | 82.25 | 21.01 | 53.22 | 78.98 | 65.42 | 18.00 | 68.80 | 18.80 |
| Qwen2.5-VL-7B-Instruct | 13.04 | 79.35 | 15.58 | 55.59 | 69.49 | 71.19 | 12.40 | 46.40 | 12.40 |
| Qwen2.5-VL-72B-Instruct | 15.94 | 75.36 | 18.12 | 45.08 | 69.83 | 60.68 | 13.60 | 48.80 | 13.60 |
| InternVL3-8B-Instruct | 12.32 | 85.51 | 14.13 | 52.50 | 67.46 | 66.78 | 12.80 | 60.40 | 13.40 |
| InternVL3-78B-Instruct | 18.84 | 87.68 | 20.29 | 62.71 | 80.00 | 74.92 | 21.60 | 69.60 | 24.40 |
| llava-hf/llava-1.5-13b-hf | 11.96 | 79.71 | 15.58 | 48.47 | 68.81 | 59.66 | 14.80 | 62.80 | 14.80 |
| Kimi-VL-A3B-Thinking-2506 | 15.94 | 75.36 | 17.75 | 54.92 | 77.24 | 66.44 | 12.00 | 55.60 | 12.40 |
| Phi-4-multimodal-instruct | 11.64 | 78.91 | 14.18 | 53.22 | 74.19 | 64.75 | 13.20 | 54.40 | 13.80 |
| Idefics3-8B-Llama3 | 8.73 | 57.45 | 13.09 | 55.25 | 67.80 | 67.80 | 14.40 | 59.60 | 14.80 |
| Gemma-3-27b-it | 14.13 | 72.83 | 17.03 | 47.80 | 72.88 | 66.44 | 20.40 | 66.80 | 22.40 |
| Ovis2-34B | 15.58 | 84.06 | 18.48 | 49.83 | 75.25 | 68.81 | 20.80 | 68.00 | 21.60 |
| human | 36.67 | 73.33 | 36.67 | 76.67 | 83.33 | 76.67 | 43.33 | 50.00 | 43.33 |

We conducted evaluations on other benchmarks, including IllusionVQA Shahgir et al. (2024) and HallusionBench Guan et al. (2024), which also contain color ambiguity. The results shown in Table 13 indicate that most VLMs struggle with this type of ambiguity, and that AmbiBench presents an even greater challenge.

Table 13: Performance of 12 VLMs on color-based ambiguities across multiple benchmarks.

| Model | IllusionVQA | HallusionBench | Ours |
|---|---|---|---|
| Gemini 2.5 Pro | 87.50 | 50.00 | 46.78 |
| GPT5 | 93.75 | 55.56 | 53.22 |
| Qwen2.5-VL-7B-Instruct | 31.25 | 55.56 | 55.59 |
| Qwen2.5-VL-72B-Instruct | 31.25 | 63.89 | 45.08 |
| InternVL3-8B-Instruct | 31.25 | 63.89 | 52.50 |
| InternVL3-78B-Instruct | 56.25 | 75.00 | 62.71 |
| llava-hf/llava-1.5-13b-hf | 56.25 | 58.33 | 48.47 |
| Kimi-VL-A3B-Thinking-2506 | 50.00 | 61.11 | 54.92 |
| Phi-4-multimodal-instruct | 56.25 | 75.00 | 53.22 |
| Idefics3-8B-Llama3 | 25.00 | 58.33 | 55.25 |
| Gemma-3-27b-it | 18.75 | 55.56 | 47.80 |
| Ovis2-34B | 56.25 | 75.00 | 49.83 |

## A.7 MORE ANALYSIS ABOUT MOTION AMBIGUITY

We constructed a baseline set consisting of 44 images from the COCO dataset—matching the number of motion images—whose labels indicate no motion. As shown in the Table 14, most VLMs can accurately label these static images, while some models (e.g., Phi-4-multimodal-instruct) still achieve only around 50% accuracy, essentially performing at chance level.

Furthermore, when we invert the prompt and ask whether the motion images in AmbiBench are static, some models exhibit a substantial performance drop. This asymmetry indicates that the models' responses are highly sensitive to the phrasing of the human query, suggesting that their behavior is at least partially driven by catering to human preference or expectation.

Overall, the motion ambiguity images remain challenging for existing VLMs.

## A.8 VISUALIZATION

We conducted visualization studies to improve interpretability. We use layer-wise attention extraction. Specifically, we average the attention weights (over all generated response tokens) from the

Table 14: Performance of 12 VLMs and humans on COCO images, Motion (static-prompt), and Motion (ours).

| Model | COCO image | Motion (static prompt) | Motion (ours) |
|---|---|---|---|
| Gemini 2.5 Pro | 96.00 | 16.00 | 76.00 |
| GPT5 | 100.00 | 0.00 | 48.00 |
| Qwen2.5-VL-7B-Instruct | 100.00 | 68.00 | 0.00 |
| Qwen2.5-VL-72B-Instruct | 100.00 | 76.00 | 36.00 |
| InternVL3-8B-Instruct | 100.00 | 8.00 | 8.00 |
| InternVL3-78B-Instruct | 100.00 | 72.00 | 44.00 |
| llava-hf/llava-1.5-13b-hf | 64.00 | 36.00 | 96.00 |
| Kimi-VL-A3B-Thinking-2506 | 100.00 | 12.00 | 40.00 |
| Phi-4-multimodal-instruct | 52.00 | 12.00 | 72.00 |
| Idefics3-8B-Llama3 | 96.00 | 24.00 | 8.00 |
| Gemma-3-27b-it | 64.00 | 0.00 | 76.00 |
| Ovis2-34B | 100.00 | 32.00 | 60.00 |

selected layer to each visual token, and then map those visual tokens back to image patches. As mid-level layers tend to capture visual information more faithfully Chen et al. (2025), we select the layer at half of the total layer depth (layer_num / 2) for visualization.

Figure 12 shows that Qwen2.5-VL-3B-Instruct primarily focuses on the outer edges to recognize a car, while Figure 13 indicates attention on the eyes and eyebrows to identify a smiling face. Similarly, in Figure 14 and 15 of multi-scene, the attention patterns differ when recognizing a horse versus a human.

## A.9 RESULTS ON MULTI-SCENE AND PEARSON ANALYSIS ACROSS CATEGORIES

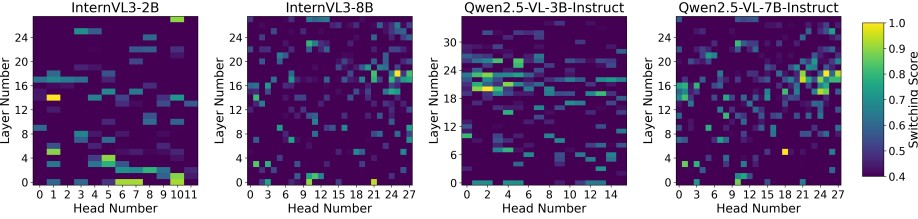

Figure 16: Perceptual-switch heads in VLMs mediate switching between two interpretations for multi-scene stimuli.

We analyze the Pearson correlation of head-activation heatmaps for the same model across bistable and multi-scene categories. The results show high consistency, further demonstrating the intrinsic nature of perception-switching heads, regardless of the data category.

Table 15: Pearson Correlation Across Models.

| Model | Pearson Correlation |
|---|---|
| Qwen2.5-VL-7B-Instruct | 0.9134 |
| Qwen2.5-VL-3B-Instruct | 0.9025 |
| InternVL3-8B | 0.9338 |
| InternVL3-2B | 0.9652 |

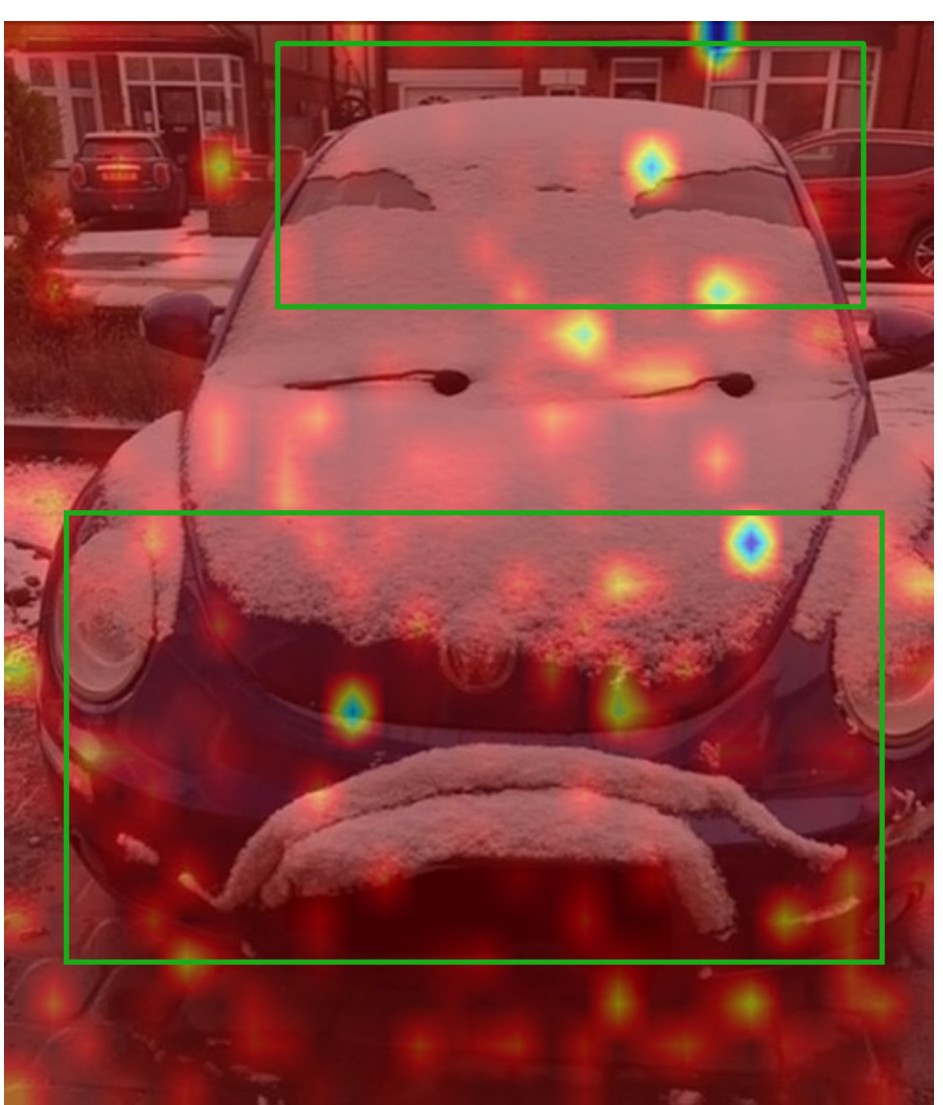

Figure 12: Bistable heatmap image before switching. Qwen2.5-VL-3B-Instruct output: Car.

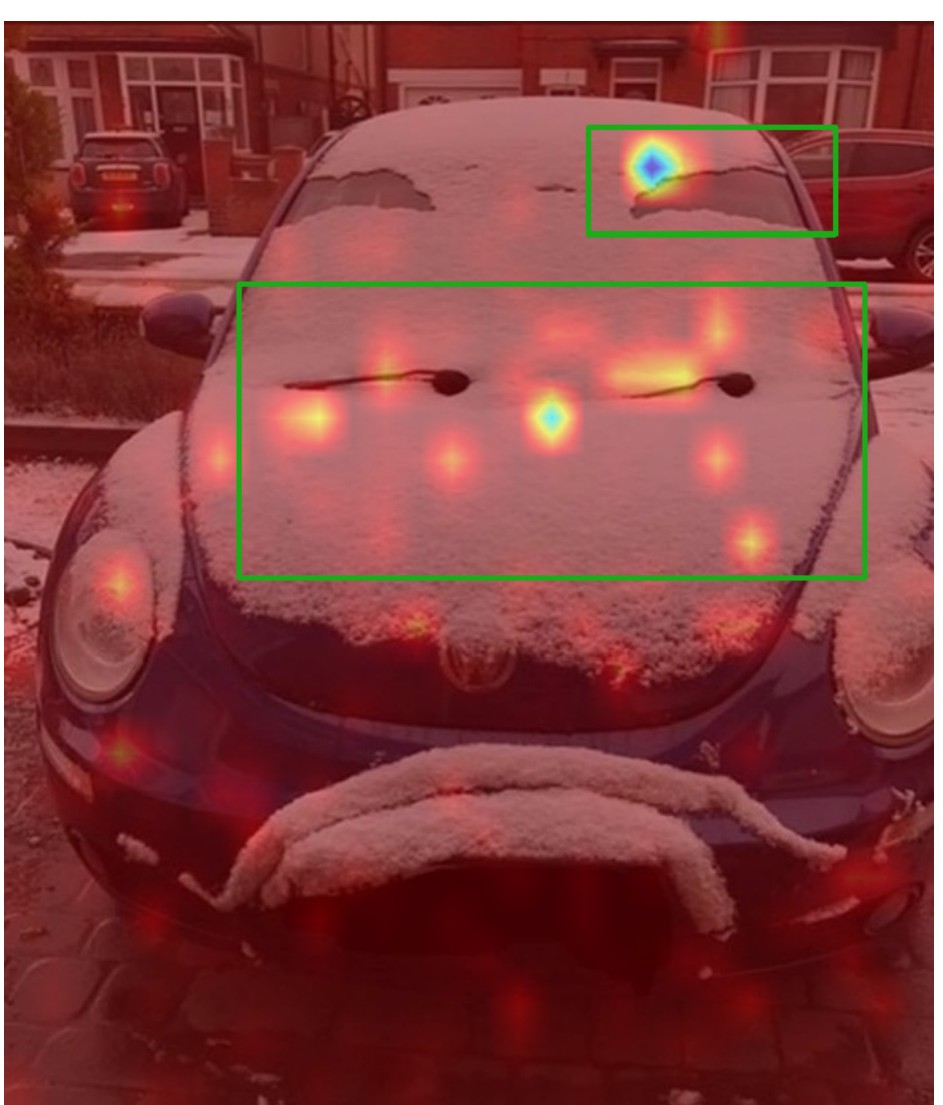

Figure 13: Bistable heatmap image after switching. Qwen2.5-VL-3B-Instruct output: Face.

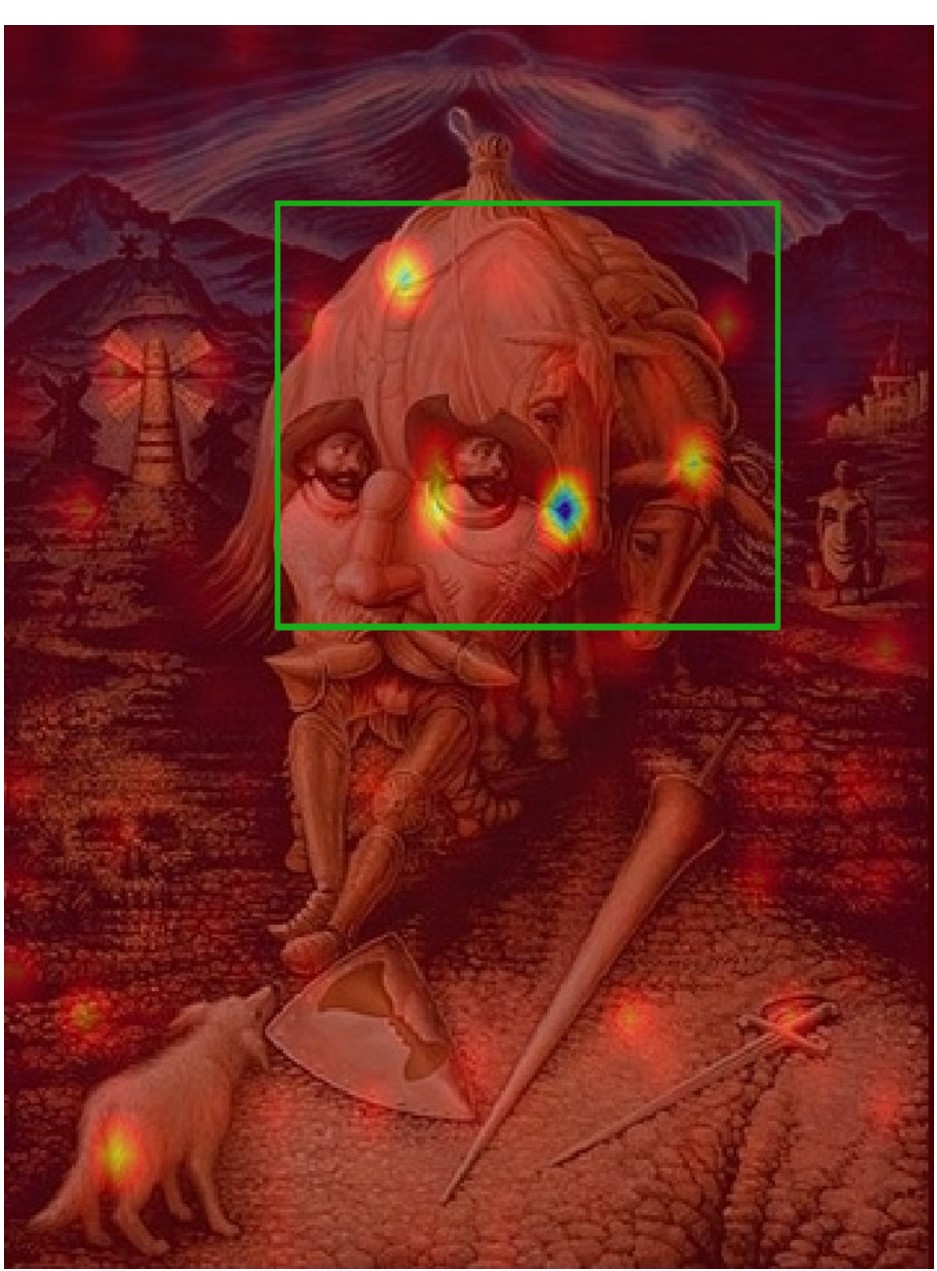

Figure 14: Multi-scene heatmap image before switching. Qwen2.5-VL-3B-Instruct output: Horse.

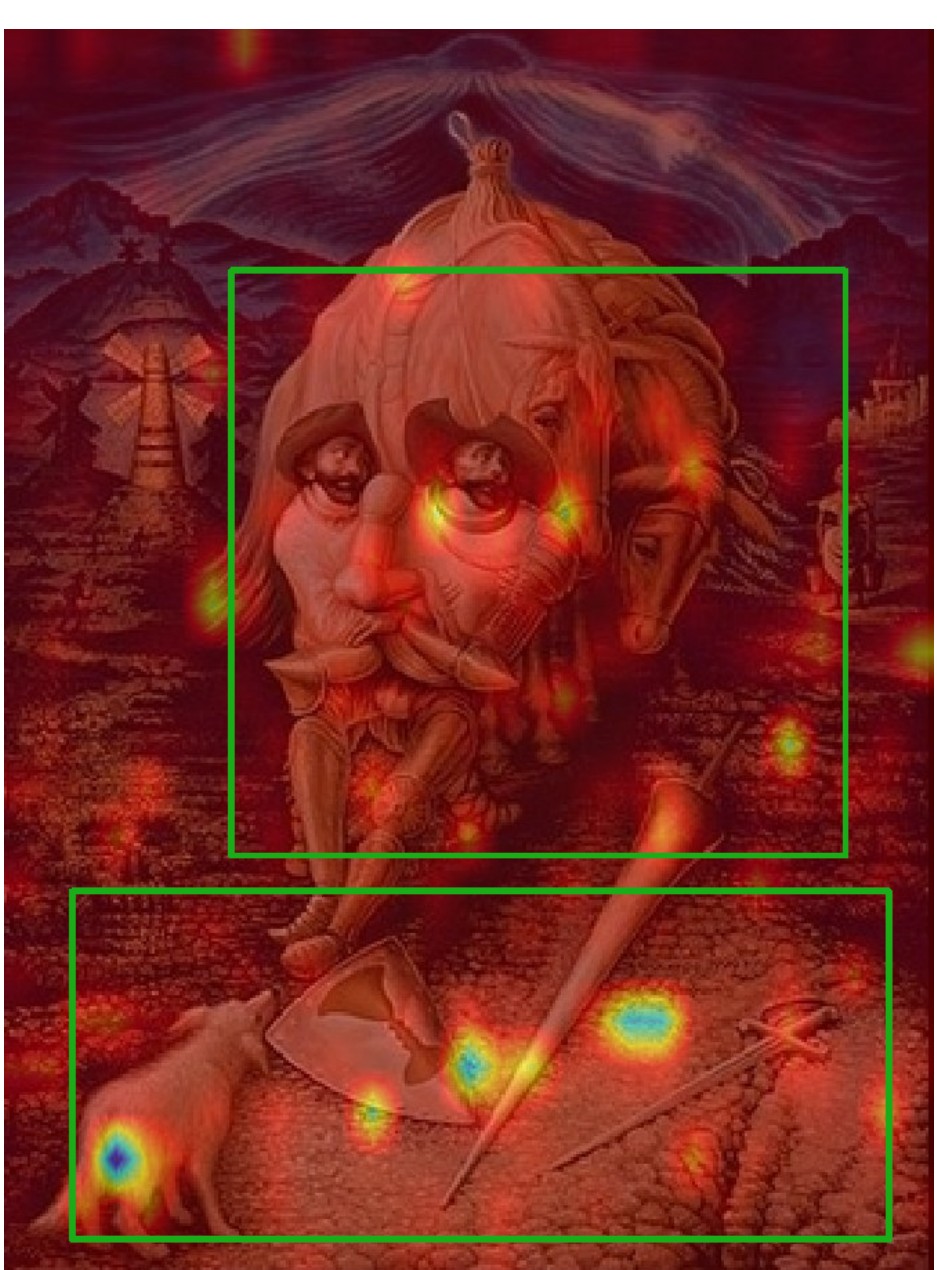

Figure 15: Multi-scene heatmap image after switching. Qwen2.5-VL-3B-Instruct output: Human.

## A.10 EXAMPLE OUTPUT OF VLM FOR DIFFERENT TASKS

**Example Outputs**

**Question Type:** Camo
**Prompt:** You are a camouflage detection expert. Carefully examine the image and locate the camouflaged object, provide the object name in one word. Output ONLY in the following format: Object: 'object name in one word'.

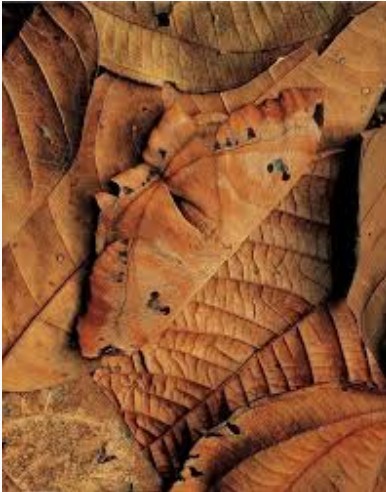

**Ground Truth: Insect**
**Correct VLM Output:** Butterfly
**Incorrect VLM Output:** Leaf (Cannot identify the camouflage object in the image.)

**Example Outputs**

**Question Type:** Bistable Choice
**Prompt:** You are given a bistable image (an image that can be perceived in more than one way). Identify which objects from the following choices are present in the image. Select **all applicable answers** by returning their **letters only**, separated by commas (e.g., A, C). If none of the objects are present, choose **E**. Choices: A: car with snow, B: face, C: Motorcycle, D: Bus, E: None Your answer:

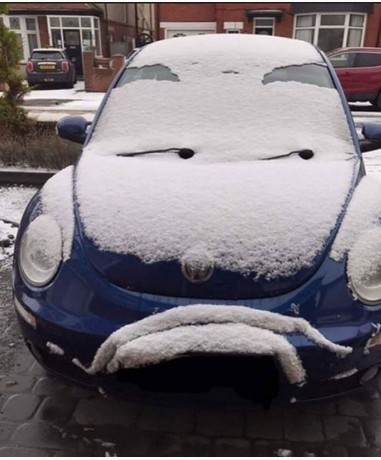

**Ground Truth: A, B**
**Correct VLM Output:** A, B
**Incorrect VLM Output:** A, C (Partially correct)
**Incorrect VLM Output:** A (Only recognize one choice)

**Example Outputs**

**Question Type:** Hybrid
**Prompt:** This is a hybrid image. What two different objects or scenes can be perceived in it, one when focusing on fine details (Object 1) and the other when viewed from a distance (Object 2)? Respond with only two object words, separated by commas, in the form of 'Object1, Object2', where Object1 is the one focusing on fine details, Object2 is the one viewing from a distance.

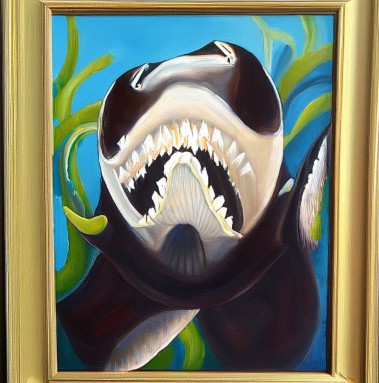

**Ground Truth: Near: A shark, Far: A panda**
**Correct VLM Output:** shark, panda
**Incorrect VLM Output:** shark, mushroom (Partially correct, failed to recognize panda)

**Example Outputs**

**Question Type:** Color
**Prompt:** This image contains two subimages with different colors. The left and right subimages depict different objects or scenes. Respond with exactly two words or two phrases, separated by a comma, in the format 'Object1, Object2', where Object1 describes the left subimage and Object2 describes the right subimage.

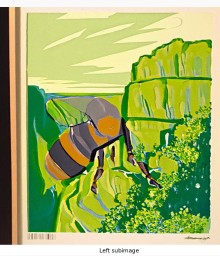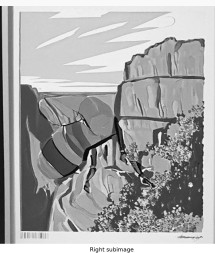

**Ground Truth: the left image looks like lithograph style, a bee, but the right image looks like lithograph style, the grand canyon.**
**Correct VLM Output:** A bee painting, Grayscale painting of a mountain
**Incorrect VLM Output:** Mountain, mountain (Same object for both left and right)

**Example Outputs**

**Question Type:** Multi-view
**Prompt:** You are given an image. Look at the image normally and de- scribe what object or scene you see. Then imagine rotating the image (90 degrees or 180 degrees, whichever makes sense) and describe what new object or scene it looks like after rotation. Respond with exactly two words or two short phrases, separated by a comma, in the format 'Object1, Object2'. Do not add extra words or explanations.

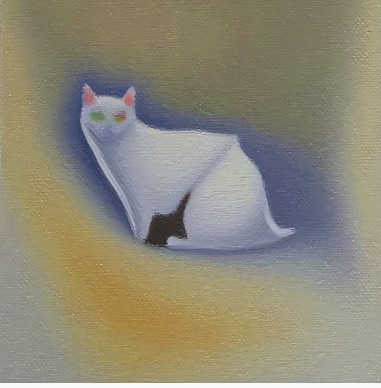

**Ground Truth: cat in a garden,bat in a cave**
**Correct VLM Output:** A white cat, a blue bat
**Incorrect VLM Output:** Cat, triangle (Partially correct)

**Example Outputs**

**Question Type:** Geometric
**Prompt:** Is the gray line in the image horizontal? The answer should be short and concise, and without explanations.

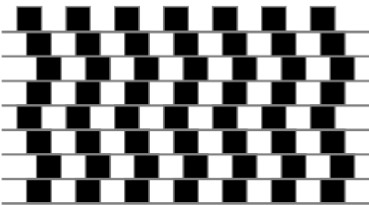

**Ground Truth: Yes**
**Correct VLM Output:** Yes
**Incorrect VLM Output:** No

**Example Outputs**

**Question Type:** Motion Image
**Prompt:** Do you think the scene in the image is moving? Respond ONLY with Yes or No.

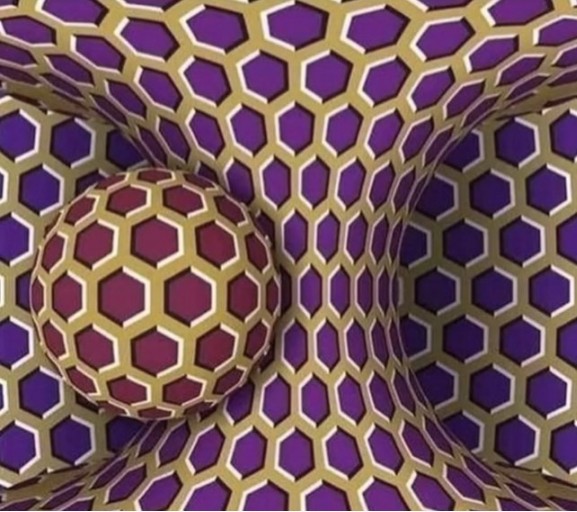

**Ground Truth: Yes**
**Correct VLM Output:** Yes
**Incorrect VLM Output:** No

**Example Outputs**

**Question Type:** Multi-scene Open
**Prompt:** You are given a multi-layer optical illusion image that contains two layers of meaning, where each layer represents a different interpretation of the same image. List all possible objects or scenes for both layers. Use short words or phrases and separate two layers with a semicolon ';'. Format your answer strictly as: 'layer1 objects; layer2 objects' Do not add any extra explanation

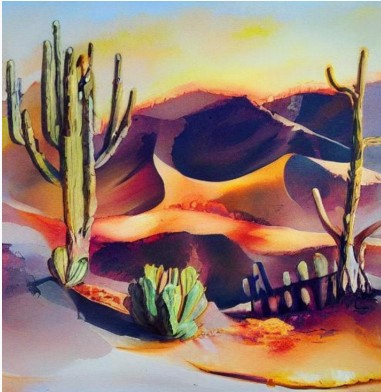

**Ground Truth: desert; skull**
**Correct VLM Output:** desert scene, sand dunes, cacti, mountains, sunset; cow skull
**Incorrect VLM Output:** desert plants; abstract shapes (Failed to recognize skull)

## Example Outputs

**Question Type:** Multi-scene Localization

**Prompt 1:** You are a object detection expert. Carefully examine the image and locate the position of all the possible skull in the image, provide the bounding box coordinates of all the elephants, the image size is Width: 802, Height: 470. If only one exists in the image, output in the following format: $'[(x\_min, y\_min, x\_max, y\_max)]'$, if multiple exist in the image, output in the following format: $'[(x\_min, y\_min, x\_max, y\_max),$ $(x\_min, y\_min, x\_max, y\_max), ...]'$, if no elephant exist in the image, return 'None'.

**Ground Truth:**

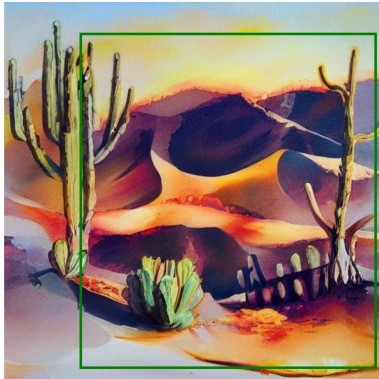

**Incorrect VLM Output:**

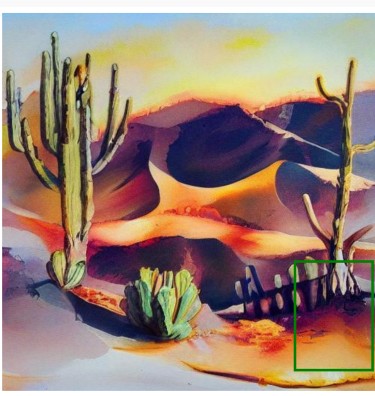

---

**Example Outputs**

**Question Type:** Multi-scene Region

**Prompt:** You are given a multi-layer optical illusion image that contains two layers of meaning in the green bounding box, where each layer represents a different interpretation of the same image. Within the **green bounding box** in the image, list all possible objects or scenes for both layers do not interpret objects outside the bounding box. Use short words or phrases and separate two layers with a semicolon ';'. Format your answer strictly as: 'layer1 objects; layer2 objects' Do not add any extra explanation

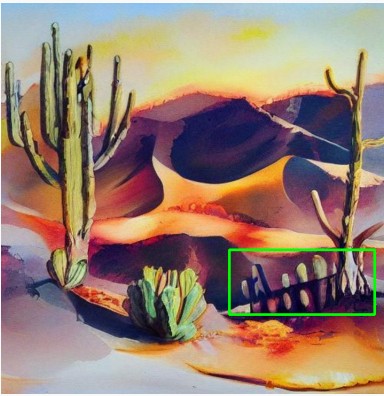

**Ground Truth: bones; teeth of skull**
**Correct VLM Output:** bones; mouth with teeth
**Incorrect VLM Output:** cactus; fence (Failed to recognize the correct objects)

---

## A.11 USE OF LLM

The LLM was primarily employed for language refinement — including polishing grammar, improving clarity, and rephrasing sentences in the manuscript.

