# OpenReview forum: "What Do VLMs See? Benchmarking Vision-Language Models on Ambiguous Images"
_ICLR.cc/2026/Conference — Submitted to ICLR 2026_

### Official Review · Reviewer_KZ8j · 2025-10-18

**Soundness:** 3
**Presentation:** 2
**Contribution:** 3
**Rating:** 4
**Confidence:** 3

**Summary:**

The paper proposes AmbiBench, a benchmark for VLMs for understanding ambiguous images compared to human interpretations. Two main categories of ambiguity are defined that include several cases of ambiguity within an image. Based on these ambiguity cases, the benchmark dataset is collected by combining existing data and purposely generated images. The benchmark demonstrates that even advanced closed-source models have difficulty interpreting ambiguous images correctly.

**Strengths:**

- The paper provides a systematic categorization of ambiguity types.
- The evaluation is extensive and includes human baseline comparisons.
- The work incorporates mechanistic interpretability through attention similarity analysis of switch heads within VLMs.

**Weaknesses:**

**W1:** The primary novelty lies in the scale and combination of several ambiguity classes together, which represents an incremental contribution in my opinion (If I understood it correctly from the paper, there are already benchmarks for the specific ambiguity classes).

**W2:** The quality of AI-generated images is not adequately addressed. If VLMs struggle to understand ambiguity, it raises questions about whether generative models can reliably design such ambiguous content.

**W3:** The dataset exhibits imbalance between categories, which may affect the validity of cross-category comparisons.

**W4:** The paper lacks comparison to existing benchmarks. Since most of these ambiguity types have been benchmarked individually in prior work, a comparison not only within this benchmark but also across existing benchmarks would provide valuable context and demonstrate added value.

**W5:** The results would benefit from more rigorous statistical analysis to make the findings more robust and meaningful.

Minor W6: Section 4.3 could be strengthened by incorporating additional mechanistic interpretability methods such as SAEs or classical saliency methods, which could visualize the switching of attention heads directly as heat maps overlaid on the images.

**Questions:**

See Weaknesses W1 to W5.

---

> ### Author Response · Authors · 2025-11-21
>
> Dear Reviewer,
>
> We thank you for the thoughtful feedback and constructive comments. We have addressed your concerns in the responses below.
>
> **Q1. The novelty of this work compared to other related works.**
>
> 1. While some works have begun to evaluate the perceptual abilities
> of VLMs on ambiguous images, existing benchmarks are limited to a narrow set of categories, small in scale, and constrained in task design. A comparison can be found in Table 4 in the Appendix. This highlights the need for a comprehensive benchmark to assess how VLMs process and reason under perceptual ambiguity.
>
> 2. Unlike recent works that mainly focus on VLM reception, our study provides a mechanistic understanding of how VLMs process both perceptual and cognitive information. The capacity of VLMs to handle cognitive ambiguity is evaluated using bistable, multi-scene, and the newly introduced mixed categories. Importantly, we also explore perceptual-switch heads in VLMs and analyze their universal, sparse, and intrinsic properties across architectures.
>
> 3. Furthermore, we added new experiments to evaluate the functional roles of the identified perceptual-switch heads, demonstrating their importance and showing that positive interventions on these heads can further improve model performance. As shown in Table 1, masking the identified perceptual-switch heads leads to a substantial performance drop across all models, whereas masking an equal number of random heads results in only marginal degradation. Table 2 shows that enhancing the activation of these heads along their corresponding functional directions improves performance of VLMs on bistable and multi-scene images.
>
> Since managing ambiguity is a higher-order aspect of human intelligence, it relates to other visual abilities such as spatial intelligence (e.g., hybrid images involve distance-dependent perceptual ambiguity) and compositional reasoning (e.g., multi-scene images require a model to integrate both local and global information, identify objects across different contexts, and flexibly switch between two plausible scene interpretations). A comprehensive benchmark is therefore essential for systematically evaluating these capabilities and advancing broader visual understanding and reasoning.
>
> | Dataset | #Images | Ambiguity Types | Modalities | Task Types | Human Study |
> |---------|---------|----------------|------------|------------|------------|
> | HallusionBench | 72 | color, geometric | Image, video | multiple-choice | Yes |
> | Grounding-Illusions  | ~100 | color, geometric | image | open-ended, multiple-choice, localization | No |
> | SemVINK  | 112 | multi-scene (single hidden) | image | multiple-choice, open-ended | No |
> | IllusionVQA | ~400 | color, multi-scene(hidden), geometric | image | multiple-choice, open-ended, soft-localization | Yes |
> | IllusionBench+ | 1051 | geometric, bistable | images | multiple-choice, open-ended | Yes |
> | **AmbiBench (Ours)** | **>2000** | **9 Types (incl. Mixed)** | Image, Video | **multiple-choice, open-ended, localization, region-based description** | yes |
>
> Table 1. Negative Intervention of different VLMs on Bistable and Multi-Scene stimuli (score: LLM-Judge Accuracy).
>
> | Models | Inter_Head | Bistable | Multi-Scene |
> |--------|------------|----------|-------------|
> | Qwen2.5-VL-3B-Instruct | random    | 38.18    | 55.26       |
> |                        | cognitive | 0.00     | 2.63        |
> | Qwen2.5-VL-7B-Instruct | random   | 41.82    | 55.26       |
> |                        | cognitive | 0.00     | 0.00        |
> | InternVL3-2B           | random   | 29.09    | 63.16       |
> |                        | cognitive | 0.00     | 0.00        |
> | InternVL3-8B           | random   | 38.18    | 63.16       |
> |                        | cognitive | 0.00     | 0.00        |
>
> Table 2. Positive Intervention of different VLMs on Bistable and Multi-Scene stimuli (score: LLM-Judge Accuracy).
>
> | Models                  | Inter_Head | Bistable | Multi-Scene |
> |-----|---|----|-----|
> | Qwen2.5-VL-3B-Instruct  | Before    | 43.64    | 57.89       |
> |                         | After     | **45.45**| **60.53**   |
> | Qwen2.5-VL-7B-Instruct  | Before    | 41.82    | 63.16       |
> |                         | After     | **45.45**| **68.42**   |
> | InternVL3-2B            | Before    | 29.09    | 63.16       |
> |                         | After     | **41.82**| **65.79**   |
> | InternVL3-8B            | Before    | 36.36    | 60.53       |
> |                         | After     | **38.18**| 60.53       |

---

> ### Author Response · Authors · 2025-11-21
>
> Thanks for your patience.
>
> **Q2. The quality of AI-generated images is not adequately addressed. If VLMs struggle to understand ambiguity, it raises questions about whether generative models can reliably design such ambiguous content.**
>
> Thank you for the insightful question. Obtaining high-quality AI-generated ambiguous images is indeed challenging. In our work, we address this concern as follows:
>
> 1. **We do not rely on VLMs to understand ambiguity in order to generate it.** Ambiguous image generating method [1] based on generative models can produce controlled visual patterns (e.g., hybrid frequency compositions, color-contrast manipulations) by following procedural rules or image-processing pipelines that do not require actual perceptual understanding. Our generation methods are grounded in well-established principles from vision science (e.g., hybrid image construction, multi-view rendering), ensuring that the ambiguous properties are structurally embedded rather than “understood” by the model.
>
> 2. **Human validation further ensures reliability.** To ensure the quality and validity of AI-generated ambiguous images, all generated samples underwent a two-stage filtering and annotation process (see Section 3.2.4 in the paper). AmbiBench retains only those images on which three independent annotators reached full agreement. For example, only 277 hybrid images are retained in Ambibench from over 2000 images.
>
> [1] Geng, Daniel, Inbum Park, and Andrew Owens. "Visual anagrams: Generating multi-view optical illusions with diffusion models." Proceedings of the IEEE/CVF Conference on Computer Vision and Pattern Recognition. 2024.
>
> **Q3. The dataset exhibits imbalance between categories, which may affect the validity of cross-category comparisons.**
>
> Obtaining high-quality ambiguous images, particularly for categories such as multi-scene and mixed, is inherently challenging. Since our main goal is evaluation rather than training, we prioritize image quality over quantity. In cross-category comparisons, we observe that all models consistently perform much worse on categories like multi-scene and mixed compared to categories such as camouflaged and geometric, often showing drops of several tens of percentage points. This substantial gap is stable across models and reflects the intrinsic difficulty of these categories rather than being an artifact of dataset imbalance.
>
> The number of images naturally decreases as the level of ambiguity increases—for example, the mixed category contains fewer samples than the multi-scene category, which in turn has fewer samples than object-level categories. This imbalance is inherent to ambiguous image construction and partially reflects the real-world distribution of such phenomena, where highly ambiguous or multi-interpretation images are far less common than object-level ambiguous cases.
>
> Finally, although AmbiBench is designed purely for evaluation, uneven category distributions may cause certain model weaknesses to appear more or less pronounced depending on sample count. We will clarify this limitation in the revised manuscript.
>
> **Q4. The paper lacks comparison to existing benchmarks. Since most of these ambiguity types have been benchmarked individually in prior work, a comparison not only within this benchmark but also across existing benchmarks would provide valuable context and demonstrate added value.**
>
> We conducted evaluations on other benchmarks, including IllusionVQA and HallusionBench, which also contain color ambiguity. The results shown in the following table indicate that most VLMs struggle with this type of ambiguity, and that AmbiBench presents an even greater challenge. We will incorporate this analysis into the revised version.
>
> Table. Performance of 12 VLMs on color-based ambiguities across multiple benchmarks.
> | Model                       | IllusionVQA | HallusionBench | Ours   |
> |------------------------------|-------------|----------------|--------|
> | Gemini 2.5 Pro               | 87.50       | 50.00          | 46.78  |
> | GPT5                         | 93.75       | 55.56          | 53.22  |
> | Qwen2.5-VL-7B-Instruct       | 31.25       | 55.56          | 55.59  |
> | Qwen2.5-VL-72B-Instruct      | 31.25       | 63.89          | 45.08  |
> | InternVL3-8B-Instruct        | 31.25       | 63.89          | 52.50  |
> | InternVL3-78B-Instruct       | 56.25       | 75.00          | 62.71  |
> | llava-hf/llava-1.5-13b-hf    | 56.25       | 58.33          | 48.47  |
> | Kimi-VL-A3B-Thinking-2506    | 50.00       | 61.11          | 54.92  |
> | Phi-4-multimodal-instruct    | 56.25       | 75.00          | 53.22  |
> | Idefics3-8B-Llama3           | 25.00       | 58.33          | 55.25  |
> | Gemma-3-27b-it               | 18.75       | 55.56          | 47.80  |
> | Ovis2-34B                    | 56.25       | 75.00          | 49.83  |

---

> > ### Author Response · Authors · 2025-11-21
> >
> > Thanks for your patience.
> >
> > **Q5. The results would benefit from more rigorous statistical analysis to make the findings more robust and meaningful.**
> >
> > The results in Table 1 and Table 2 in the paper show that the performance differences between humans and VLMs are substantial, with gaps of several tens of percentage points in some categories. These differences are robust and clearly highlight the limitations of VLMs in handling ambiguous visual information.
> >
> > Moreover, we compute the average performance of VLMs across all object-level categories, providing a comprehensive summary of model capabilities and highlighting consistent trends in their relative strengths and weaknesses. Notably, Gemini and GPT-5 models perform relatively better, and these analyses demonstrate that our evaluation includes basic statistical considerations to support the reported findings. We are happy to provide additional analyses if the reviewer have further concerns.
> >
> > **Q6. Section 4.3 could be strengthened by incorporating additional mechanistic interpretability methods such as SAEs or classical saliency methods, which could visualize the switching of attention heads directly as heat maps overlaid on the images.**
> >
> > Following your suggestions, we conducted visualization studies to improve interpretability. Figure 8 ([Rebuttal PDF](https://anonymous.4open.science/r/ICLR_ambibench_anoymous-12A4/Ambi_Benchmark__ICLR_2026_Response_.pdf)) shows that Qwen2.5-VL-3B-Instruct primarily focuses on the outer edges to recognize a car, while Figure 9 indicates attention on the eyes and eyebrows to identify a smiling face. Similarly, in multi-scene images, the attention patterns differ when recognizing a horse versus a human (Figure 10 and 11 at [Rebuttal PDF](https://anonymous.4open.science/r/ICLR_ambibench_anoymous-12A4/Ambi_Benchmark__ICLR_2026_Response_.pdf)).

---

> > > ### Comment · Reviewer_KZ8j · 2025-11-22
> > >
> > > Dear Authors,
> > >
> > > thank you for the extensive rebuttal. I see Q1 -Q4 adequately addressed.
> > >
> > > Regarding Q5: Claiming a result is "robust" only due to a large difference in percentage points is inadequate. A notion of robustness is not only about the positions within a ranking but also about the consistency of the results themselves. A statistical analysis should quantify how consistent/stable a result is (e.g. confidence intervals) and how uncertain a method is about its estimate. This gives important context to the ranking such as whether observed differences are meaningful or within the margin of error, which methods show reliable performance across multiple runs versus those with high variance, and how much confidence we should have when choosing one method over another based on the available evidence.
> > >
> > > Regarding Q6: What XAI methods were used here? Is this just raw last layer attention visualisation? The saliency maps look like they show single patches/tokens and do not really focus on specific features within the images. Is this due to the used method or is the method faithful and the model really uses only a small subset of tokens (would indicate maybe overfitting or similar?)

---

> ### Author Response · Authors · 2025-11-25
>
> Dear Reviewer,
>
> We sincerely thank the reviewer for their thoughtful comments and prompt response. We are pleased that our previous reply helped clarify some of your concerns. Below, we provide a more detailed response to the remaining points:
>
> **Q. More statistical analysis for robustness**
>
> To ensure the stability and reproducibility of our experimental results, we configure the VLMs in a deterministic mode (i.e., False for stochastic sampling), following standard benchmarking practices for fairness.
>
> We additionally evaluate each model under its original sampling settings (temperature = 1) and repeat the experiments five times with different random seeds. The results are summarized in the following table. We evaluate performance across object-level categories (color, multi-view, and hybrid), as well as scene-level and mixed-level categories. The overall trends are consistent with those reported in Tables 1 and 2 of the main paper.
>
> A t-test comparing performance across these categories confirms that our findings—specifically, that VLMs perform worse on scene-level and mixed categories than on object-level categories, and that performance on color is higher than on multi-view and hybrid—are statistically significant (p-value$\ll$0.05).
>
> Table1. Performance of 12 VLMs on different categories (mean and variant over 5 seeds).
> | Model                        | Multi-Scene (open) | Mix (open) | Color            | Multi-View      | Hybrid          |
> |-|-|-|-|-|-|
> | Gemini 2.5 Pro               | 63.68 ± 7.33     | 1.43 ± 1.75 | 48.27 ± 1.19    | 19.76 ± 1.23   | 18.33 ± 0.96   |
> | GPT5                         | 74.74 ± 2.11     | 1.43 ± 3.19 | 51.59 ± 0.92    | 16.24 ± 1.20   | 20.58 ± 1.12   |
> | Qwen2.5-VL-7B-Instruct       | 26.32 ± 0.00     | 0.00 ± 0.00 | 47.53 ± 4.55    | 9.44 ± 1.49    | 9.15 ± 2.31    |
> | Qwen2.5-VL-72B-Instruct      | 37.89 ± 2.11     | 0.00 ± 0.00 | 41.49 ± 2.31    | 9.76 ± 1.92    | 13.19 ± 1.74   |
> | InternVL3-8B-Instruct        | 23.16 ± 4.21     | 0.00 ± 0.00 | 49.36 ± 1.76    | 9.20 ± 1.81    | 11.34 ± 0.69   |
> | InternVL3-78B-Instruct       | 38.95 ± 1.05     | 0.00 ± 0.00 | 57.97 ± 2.39    | 16.48 ± 2.56   | 14.53 ± 2.15   |
> | llava-hf/llava-1.5-13b-hf    | 21.05 ± 4.08     | 0.00 ± 0.00 | 30.89 ± 8.92    | 12.56 ± 1.12   | 10.08 ± 1.12   |
> | Kimi-VL-A3B-Thinking-2506    | 32.63 ± 2.11     | 0.00 ± 0.00 | 48.63 ± 1.59    | 11.52 ± 1.13   | 14.83 ± 0.72   |
> | Phi-4-multimodal-instruct    | 12.63 ± 4.21     | 0.00 ± 0.00 | 46.92 ± 3.66    | 12.56 ± 0.32   | 10.36 ± 0.64   |
> | Idefics3-8B-Llama3           | 27.37 ± 1.29     | 0.00 ± 0.00 | 48.47 ± 3.39    | 10.08 ± 2.08   | 7.90 ± 0.41    |
> | Gemma-3-27b-it               | 54.74 ± 1.05     | 0.00 ± 0.00 | 49.29 ± 0.41    | 19.44 ± 0.48   | 15.07 ± 0.29   |
> | Ovis2-34B                    | 26.84 ± 4.21     | 0.00 ± 0.00 | 51.25 ± 0.14    | 17.28 ± 1.76   | 16.23 ± 0.14   |

---

> ### Author Response · Authors · 2025-11-25
>
> Thanks for your patience.
>
> Table 2. P-values of paired comparisons across 12 VLMs.
> | Model  | Color vs Multi-View   | Color vs Hybrid      | Color vs Multi-Scene | Color vs Mix        | Multi-View vs Multi-Scene | Multi-View vs Mix   | Hybrid vs Multi-Scene | Hybrid vs Mix      |
> |--|-|-|-|-|-|-|-|-|
> | Gemini 2.5 Pro       | 1.54×10^-11         | 1.11×10^-11       | 1.42×10^-06        | 1.06×10^-22       | 2.13×10^-10             | 6.44×10^-12       | 1.59×10^-08         | 1.27×10^-09       |
> | GPT5                 | 1.94×10^-10         | 1.15×10^-11       | 4.48×10^-12        | 7.14×10^-20       | 6.10×10^-11             | 4.90×10^-12       | 1.79×10^-10         | 2.13×10^-05       |
> | Qwen2.5-VL-7B        | 3.37×10^-05         | 4.77×10^-05       | 3.94×10^-01        | 2.52×10^-06       | 7.94×10^-05             | 6.23×10^-07       | 1.05×10^-03         | 4.77×10^-04       |
> | Qwen2.5-VL-72B       | 7.66×10^-06         | 1.58×10^-05       | 1.68×10^-02        | 1.98×10^-07       | 5.10×10^-06             | 2.51×10^-10       | 3.91×10^-04         | 4.01×10^-05       |
> | InternVL3-8B         | 9.43×10^-08         | 1.08×10^-07       | 2.65×10^-03        | 3.44×10^-09       | 8.31×10^-07             | 3.67×10^-08       | 3.27×10^-05         | 1.36×10^-06       |
> | InternVL3-78B        | 1.84×10^-08         | 6.94×10^-09       | 2.60×10^-07        | 1.91×10^-10       | 2.92×10^-06             | 3.94×10^-08       | 1.23×10^-05         | 4.51×10^-04       |
> | LLaVA-1.5-13B        | 3.20×10^-03         | 3.55×10^-03       | 6.84×10^-02        | 2.59×10^-04       | 1.22×10^-02             | 1.22×10^-04       | 3.29×10^-02         | 4.62×10^-06       |
> | Kimi-VL              | 2.41×10^-06         | 3.93×10^-06       | 2.58×10^-03        | 1.44×10^-15       | 3.47×10^-08             | 4.13×10^-06       | 2.12×10^-04         | 5.25×10^-08       |
> | Phi-4-MM             | 5.26×10^-05         | 5.26×10^-05       | 2.15×10^-01        | 9.70×10^-07       | 4.46×10^-02             | 2.09×10^-06       | 7.92×10^-03         | 2.29×10^-06       |
> | Idefics3-8B          | 4.50×10^-09         | 4.15×10^-09       | 9.61×10^-03        | 1.22×10^-13       | 3.90×10^-08             | 1.24×10^-06       | 3.29×10^-06         | 1.42×10^-06       |
> | Gemma-3-27B          | 1.73×10^-13         | 2.42×10^-13       | 7.59×10^-10        | 3.38×10^-16       | 2.89×10^-12             | 2.05×10^-09       | 1.10×10^-10         | 1.71×10^-08       |
> | Ovis2-34B            | 5.21×10^-07         | 1.79×10^-06       | 2.67×10^-03        | 2.81×10^-11       | 4.08×10^-08             | 9.67×10^-08       | 6.74×10^-05         | 8.65×10^-10       |
>
> **Q. Further details on the visualization**
>
> The method we employ is layer-wise attention extraction. Concretely, for a chosen layer, we average the attention weights from all generated response tokens to each visual token, and then map these visual tokens back to their corresponding image patches. This procedure follows the methodology described in [1].
>
> The reason the saliency maps sometimes appear to highlight only a few patches is that the attention over visual tokens varies significantly. After normalization, a small number of tokens with strong salience remain visible, while others are suppressed. Nonetheless, as shown in Figure 8 (car label), we can also observe clusters of highlighted patches in front of the car. As also noted in [1], mid-level layers tend to produce more faithful visual grounding than early or late layers. Therefore, for visualization we select the layer at half of the model’s depth (i.e., layer\_num/2), where attention patterns are typically the most interpretable and semantically meaningful.
>
> We will clarify these methodological details and the rationale behind the layer choice in the revision.
>
> [1]: Why Is Spatial Reasoning Hard for VLMs? An Attention Mechanism Perspective on Focus Areas
>
> We hope we have addressed all your concerns and would be glad to discuss any remaining questions you may have.
>
> Best regards,
>
> The Authors

---

> ### Comment · Reviewer_KZ8j · 2025-11-25
>
> Dear Authors,
>
> thank you for your clarifications and additional experiments. I updated my score accordingly.
>
> As a comment: I think you would have to correct the p-values in Table 2 for [multiple-testing](https://en.wikipedia.org/wiki/Multiple_comparisons_problem) as you always have the same H0 Hypothesis in the tests.

---

> ### Author Response · Authors · 2025-11-27
>
> Dear Reviewer,
>
> We sincerely thank you for your careful reading and constructive comments, and we are glad that the additional experiments have addressed your concerns.
>
> Regarding your comment on multiple testing, we provide the following analysis (correct the p-values) for object-level ambiguity (multiple categories) vs scene-level ambiguity, and object-level ambiguity vs mixed level ambiguity. The small p-values of the following Table 1 indicate that VLMs perform worse on scene-level and mixed categories compared to object-level categories.
>
> Meanwhile, we compare GPT-5 and Gemini 2.5 Pro with Qwen2.5-VL-7B-Instruct using multiple-testing correction across several categories. The results from Table 2 show that for four categories, both GPT-5 and Gemini 2.5 Pro outperform Qwen2.5-VL-7B with statistical significance. However, for the more challenging mixed category, the performance differences between models are not statistically significant.
>
> Table 1. Bonferroni-corrected p-values (group size = 3 per set).
>
> | **Model**         | **Color vs Multi-Scene** | **Multi-View vs Multi-Scene** | **Hybrid vs Multi-Scene** | **Color vs Mix** | **Multi-View vs Mix** | **Hybrid vs Mix** |
> |-------------------|--------------------------|-------------------------------|----------------------------|-------------------|------------------------|--------------------|
> | Gemini 2.5 Pro    | 4.26e-06                 | 6.39e-10                      | 4.77e-08                  | 3.18e-22          | 1.93e-11               | 3.81e-09           |
> | GPT5              | 1.34e-11                 | 1.83e-10                      | 5.37e-10                  | 2.14e-19          | 1.47e-11               | 6.39e-05           |
> | Qwen2.5-VL-7B     | 3.94e-01                 | 2.38e-04                      | 3.15e-03                  | 7.56e-06          | 1.87e-06               | 1.43e-03           |
> | Qwen2.5-VL-72B    | 5.04e-02                 | 1.53e-05                      | 1.17e-03                  | 5.94e-07          | 7.53e-10               | 1.20e-04           |
> | InternVL3-8B      | 7.95e-03                 | 2.49e-06                      | 9.81e-05                  | 1.03e-08          | 1.10e-07               | 4.08e-06           |
> | InternVL3-78B     | 7.80e-07                 | 8.76e-06                      | 3.69e-05                  | 5.73e-10          | 1.18e-07               | 1.35e-03           |
> | LLaVA-1.5-13B     | 2.05e-01                 | 3.66e-02                      | 9.87e-02                  | 7.77e-04          | 3.66e-04               | 1.39e-05           |
> | Kimi-VL           | 7.74e-03                 | 1.04e-07                      | 6.36e-04                  | 4.32e-15          | 1.24e-05               | 1.58e-07           |
> | Phi-4-MM          | 6.44e-01                 | 1.34e-01                      | 2.38e-02                  | 2.91e-06          | 6.27e-06               | 6.87e-06           |
> | Idefics3-8B       | 2.88e-02                 | 1.17e-07                      | 9.87e-06                  | 3.66e-13          | 3.72e-06               | 4.26e-06           |
> | Gemma-3-27B       | 2.28e-09                 | 8.67e-12                      | 3.30e-10                  | 1.01e-15          | 6.15e-09               | 5.13e-08           |
> | Ovis2-34B         | 7.99e-03                 | 1.22e-07                      | 2.02e-04                  | 8.43e-11          | 2.90e-07               | 2.59e-09           |
>
> Table 2. Bonferroni-corrected p-values of comparing GPT5 / Gemini 2.5 Pro against Qwen2.5-VL-72B-Instruct.
> | Model          | Multi-Scene | Mix    | Color    | Multi-View | Hybrid   |
> |----------------|-------------|--------|----------|------------|----------|
> | GPT5           | 0.000       | 1.865  | 0.00106  | 0.002185   | 0.000525 |
> | Gemini2.5-Pro  | 0.004365    | 0.71   | 0.00565  | 0.000145   | 0.00515  |
>
> Best regards,
>
> The Authors

---

### Official Review · Reviewer_GvwZ · 2025-10-19

**Soundness:** 2
**Presentation:** 2
**Contribution:** 2
**Rating:** 2
**Confidence:** 4

**Summary:**

The paper introduces AmbiBench, designed to evaluate vision-language models (VLMs) on ambiguous imagery. The benchmark covers three major ambiguity levels, object-level, scene-level, and mixed, and includes 2,687 carefully curated Q&A pairs across nine categories. There are 4 types of tasks: open-ended, multiple-choice, ambiguous localization, and local region description. Experimental results show large performance gaps between humans and VLMs. Furthermore, by comparing attention head activations before and after informing models that an image is bistable, the authors compute a switching score based on cosine similarity and identify a small subset of attention heads that act as perceptual-switch heads, suggesting an emergent internal mechanism for perceptual switching within VLMs.

**Strengths:**

- The paper presents a well-structured benchmark that systematically covers multiple levels of visual ambiguity and diverse task types.
- The authors evaluate a wide range of VLMs and conduct comparisons with human performance, offering insight into the gap between machine and human perceptual reasoning.

**Weaknesses:**

- While AmbiBench covers a broader range of ambiguity types, it remains unclear what makes it distinctively different from prior illusion-based benchmarks. For example, are there cases where models perform well on IllusionVQA but poorly on AmbiBench? If so, a discussion on what specific perceptual or reasoning abilities the models lack would strengthen the paper.
- The paper highlights that all evaluated VLMs perform significantly worse than humans, but provides no direction on how such performance gaps might be closed. Suggesting possible strategies or training paradigms would make the contribution more forward-looking rather than purely diagnostic.
- The paper repeatedly emphasizes that AmbiBench includes difficult tasks, yet does not articulate why succeeding on these tasks is meaningful. It would be valuable to discuss whether improvement on these ambiguous tasks correlates with broader visual reasoning or compositional understanding in other domains.
- The analysis in Sec. 4.3 relies solely on cosine similarity between attention activations before and after revealing bistability, which suggests correlation rather than causation. It remains unverified whether the model truly performs perceptual switching between interpretations, or if the observed activation change simply reflects prompt-induced attention shifts.

**Questions:**

- Since part of the dataset was collected from online sources, is there a possibility that some images were previously seen during pretraining, especially by frontier models such as GPT-5 or Gemini? If so, how was data contamination mitigated?
- IllusionVQA intentionally filters out images that GPT-4V already explains well to retain only challenging cases. Was any similar filtering or difficulty calibration applied in AmbiBench to ensure that the benchmark indeed focuses on ambiguous or failure-prone examples?

---

> ### Author Response · Authors · 2025-11-21
>
> Dear Reviewer,
>
> We deeply appreciate the time and effort you have dedicated to reviewing our work. Below, we address the questions you raised.
>
> **Q1. While AmbiBench covers a broader range of ambiguity types, it remains unclear what makes it distinctively different from prior illusion-based benchmarks.**
>
> 1. Existing benchmarks are **limited in categories, scale, and task designs** (see Table 4 in the Appendix for comparison).
>
> The existing benchmarks focus primarily on color and geometric illusions, emphasizing VLM perception. In contrast, AmbiBench provides a mechanistic understanding of how VLMs process both **perceptual and cognitive information**. We evaluate VLMs’ ability to handle cognitive ambiguity through bistable, multi-scene, and newly introduced mixed categories. Importantly, we identify and **analyze perceptual-switch heads in VLMs**, revealing their sparse, universal, and intrinsic properties across architectures.
>
> We also conducted evaluations on other benchmarks, including IllusionVQA and HallusionBench, which also contain color ambiguity. The results shown in the following table indicate that AmbiBench presents an even greater challenge.
>
> AmbiBench also implements **comprehensive evaluation protocols**, including a suite of tasks that assess ambiguous visual understanding and fine-grained local-to-global reasoning.
>
> More categories and comprehensive tasks help to **understand VLMs' performance across different categories and tasks**. For example, compared to camouflaged, color, and geometric images, their performance drops sharply on bistable, hybrid, and multi-view images. Further analysis can be found in the section "Performance Across Different Categories and Tasks" in the paper.
>
> Table. Performance of 12 VLMs on color-based ambiguities across multiple benchmarks.
> | Model                       | IllusionVQA | HallusionBench | Ours   |
> |------------------------------|-------------|----------------|--------|
> | Gemini 2.5 Pro               | 87.50       | 50.00          | 46.78  |
> | GPT5                         | 93.75       | 55.56          | 53.22  |
> | Qwen2.5-VL-7B-Instruct       | 31.25       | 55.56          | 55.59  |
> | Qwen2.5-VL-72B-Instruct      | 31.25       | 63.89          | 45.08  |
> | InternVL3-8B-Instruct        | 31.25       | 63.89          | 52.50  |
> | InternVL3-78B-Instruct       | 56.25       | 75.00          | 62.71  |
> | llava-hf/llava-1.5-13b-hf    | 56.25       | 58.33          | 48.47  |
> | Kimi-VL-A3B-Thinking-2506    | 50.00       | 61.11          | 54.92  |
> | Phi-4-multimodal-instruct    | 56.25       | 75.00          | 53.22  |
> | Idefics3-8B-Llama3           | 25.00       | 58.33          | 55.25  |
> | Gemma-3-27b-it               | 18.75       | 55.56          | 47.80  |
> | Ovis2-34B                    | 56.25       | 75.00          | 49.83  |
>
> 2. Existing benchmarks are primarily collected from online sources due to the challenges of generating ambiguous images. Our work introduces methods to generate additional ambiguous images, including a new mixed category. This approach not only expands the diversity of ambiguous stimuli but also helps mitigate potential data contamination.
>
> 3. Additionally, we have conducted new experiments to evaluate the functional roles of the identified perceptual-switch heads, demonstrating their importance and showing that positive interventions on these heads can further improve model performance. As shown in Table 1, masking the identified perceptual-switch heads leads to a substantial performance drop across all models, whereas masking an equal number of random heads results in only marginal degradation. Table 2 shows that enhancing the activation of these heads along their corresponding functional directions improves performance of VLMs on bistable and multi-scene images.
>
> Table 1. Negative Intervention of different VLMs on Bistable and Multi-Scene stimuli (score: LLM-Judge Accuracy).
>
> | Models | Inter_Head | Bistable | Multi-Scene |
> |-|--|--|--|
> | Qwen2.5-VL-3B-Instruct | random    | 38.18    | 55.26       |
> |  | cognitive | 0.00     | 2.63        |
> | Qwen2.5-VL-7B-Instruct | random   | 41.82    | 55.26       |
> |    | cognitive | 0.00     | 0.00        |
> | InternVL3-2B           | random   | 29.09    | 63.16       |
> |      | cognitive | 0.00     | 0.00        |
> | InternVL3-8B           | random   | 38.18    | 63.16       |
> |    | cognitive | 0.00     | 0.00        |
>
> Table 2. Positive Intervention of different VLMs on Bistable and Multi-Scene stimuli (score: LLM-Judge Accuracy).
>
> | Models  | Inter_Head | Bistable | Multi-Scene |
> |-|---|--|-|
> | Qwen2.5-VL-3B-Instruct  | Before    | 43.64    | 57.89       |
> |  | After     | **45.45**| **60.53**   |
> | Qwen2.5-VL-7B-Instruct  | Before    | 41.82    | 63.16|
> |    | After     | **45.45**| **68.42**   |
> | InternVL3-2B  | Before    | 29.09    | 63.16|
> |   | After     | **41.82**| **65.79**   |
> | InternVL3-8B   | Before    | 36.36    | 60.53 |
> |  | After     | **38.18**| 60.53       |

---

> ### Author Response · Authors · 2025-11-21
>
> Thanks for your patience.
>
> **Q2. The paper highlights that all evaluated VLMs perform significantly worse than humans, but provides no direction on how such performance gaps might be closed. Suggesting possible strategies or training paradigms would make the contribution more forward-looking rather than purely diagnostic.**
>
> Based on our analysis of perceptual-switch heads in VLMs, we propose a positive intervention strategy, where these heads are guided toward the correct interpretation. Our experiments (Table 2 in response to Q1) show that such interventions can improve model performance on ambiguous images, suggesting a potential pathway for bridging the gap between VLMs and human perception. This points toward future directions in targeted fine-tuning or attention-guided training paradigms to enhance higher-order cognitive abilities in VLMs.
>
> **Q3. The paper repeatedly emphasizes that AmbiBench includes difficult tasks, yet does not articulate why succeeding on these tasks is meaningful. It would be valuable to discuss whether improvement on these ambiguous tasks correlates with broader visual reasoning or compositional understanding in other domains.**
>
> Thanks for the suggestion. We have discussed this in the revised manuscript.
>
> The categories in AmbiBench are clearly related to broader visual reasoning, including spatial intelligence and compositional understanding. For example, hybrid images involve distance-dependent perceptual ambiguity, and success on these types will also improve spatial intelligence.
> More challenging categories, such as multi-scene images, require a model to integrate both local and global information, identify objects across different contexts, and flexibly switch between two plausible scene interpretations. This process is highly related to compositional understanding.
>
> These ambiguous tasks also align with higher-order cognitive abilities observed in humans, including cognitive flexibility, creativity, imagination, abstract reasoning, and the ability to manage perceptual conflict or ambiguity. Therefore, improvements on AmbiBench tasks are meaningful because they are closely tied to core aspects of human intelligence.
>
> **Q4. The analysis in Sec. 4.3 relies solely on cosine similarity between attention activations before and after revealing bistability, which suggests correlation rather than causation. It remains unverified whether the model truly performs perceptual switching between interpretations, or if the observed activation change simply reflects prompt-induced attention shifts.**
>
> We conducted experiments to evaluate the importance of the identified perceptual-switch heads. Table 1, in response to Q1, shows that masking these heads leads to a significant performance drop across different models. This indicates that the identified heads play a functional role in perceptual switching rather than merely reflecting prompt-induced attention shifts.
>
> **Q5. Since part of the dataset was collected from online sources, is there a possibility that some images were previously seen during pretraining, especially by frontier models such as GPT-5 or Gemini? If so, how was data contamination mitigated?**
>
> Data contamination is discussed in line 419 and line 429 of the paper. Importantly, contamination is not specific to GPT-5 or Gemini—it is a general issue for all models. We mitigate it in the following ways:
>
> 1. **Designing effective tasks.** For example, in a multi-view image, GPT-5 may produce a fluent global caption, but when we use the local region description task, it cannot provide a correct answer. This shows that although frontier models may recall memorized descriptions, they cannot reliably reason under specifically designed tasks. The results in Table 2 further demonstrate the effectiveness of this task design in mitigating contamination.
>
> 2. **Generating new ambiguous images** for several categories. In AmbiBench, we newly generate ambiguous images for hybrid, color, multi-view, and mixed categories. This also helps reduce the risk of data contamination.
>
> **Q6. IllusionVQA intentionally filters out images that GPT-4V already explains well to retain only challenging cases. Was any similar filtering or difficulty calibration applied in AmbiBench to ensure that the benchmark indeed focuses on ambiguous or failure-prone examples?**
>
> We also consider data contamination in our paper, but we address it in different ways: by designing effective tasks and generating new ambiguous images. The details can be found in our response to Q5.
>
> Filtering images using a single model can help mitigate data contamination to some extent. However, since contamination is not specific to any one model but is a general issue affecting all models—and different models may have been exposed to different images—any online image could potentially be contaminated. Therefore, relying on a single model to filter contaminated examples does not generalize to other models.

---

### Official Review · Reviewer_CBZt · 2025-10-26

**Soundness:** 3
**Presentation:** 3
**Contribution:** 2
**Rating:** 6
**Confidence:** 2

**Summary:**

The paper focuses on the evaluation gap in visual-language models regarding their ability to perceive and reason about ambiguous images. It innovatively proposes the AmbiBench benchmark, which fills a key void in systematically assessing VLMs’ higher-order human-like cognitive abilities, such as abstract interpretation and ambiguity resolution. The construction of the AmbiBench benchmark is methodologically rigorous. It includes 2,238 ambiguous images and 2,687 visual question–answer pairs, categorized into nine types of ambiguity. The paper clearly reveals the core limitations of VLMs in processing ambiguous images.

**Strengths:**

1. The paper identifies a key gap in current VLM evaluations. While VLMs perform well on standard benchmarks, their ability in higher-order cognition, such as abstract reasoning and ambiguity resolution, remains underexplored.
2. AmbiBench includes 2,238 carefully collected and ambiguous images across nine categories (seven object-level, one scene-level, one mixed) and 2,687 curated QA pairs.
3. Four complementary tasks (open-ended QA, multiple-choice, object localization, and local region description) capture different aspects of VLM performance. They jointly assess global and local perception, providing a more comprehensive evaluation than earlier single-task benchmarks.

**Weaknesses:**

1.	The mixed ambiguity category, newly introduced as a core class, contains only 28 samples. Although these samples were manually curated, the small scale may affect the statistical stability of the evaluation results for this category. It is recommended to either include more mixed ambiguity samples or provide an explanation for the limited sample size and its potential impact on the results.
2.	The investigation of the perception-switching head mechanism focuses solely on bistable images, without further validation of its generalizability to other types of ambiguity (e.g., multi-scene or mixed ambiguity).

**Questions:**

See weaknesses.

1. The distinction from TET [1] can be further elaborated and discussed.
2. The experiments do not clearly specify the LLM judge’s criteria for verifying answers in open-ended and multiple-choice questions, particularly regarding the definition of all expected explanations. Providing additional details would enhance the reproducibility of the evaluation process.

[1] PIXELS, PATTERNS, BUT NO POETRY: TO SEE THE WORLD LIKE HUMANS, 2025.

---

> ### Author Response · Authors · 2025-11-21
>
> Dear Reviewer,
>
> Thank you for your constructive comments and suggestions. Please find our point-by-point responses to your concerns below.
>
> **Q1. The mixed ambiguity category, newly introduced as a core class, contains only 28 samples. Although these samples were manually curated, the small scale may affect the statistical stability of the evaluation results for this category. It is recommended to either include more mixed ambiguity samples or provide an explanation for the limited sample size and its potential impact on the results.**
>
> Obtaining high-quality ambiguous images, particularly for categories such as mixed ambiguity category, is inherently challenging. Since our main goal is evaluation rather than training, we prioritize image quality over quantity.
> The number of images decreases as ambiguity increases—for example, the mixed category has fewer samples than multi-scene, which in turn has fewer than object-level categories. This imbalance is inherent to ambiguous image creation and partially reflects real-world distributions, where highly ambiguous or multi-interpretation images are rarer than object-level ambiguous cases.
>
> Although AmbiBench is designed purely for evaluation, uneven category distributions may cause certain model weaknesses to appear more or less pronounced depending on sample count. We will clarify this limitation in the revised manuscript.
>
> **Q2. The investigation of the perception-switching head mechanism focuses solely on bistable images, without further validation of its generalizability to other types of ambiguity (e.g., multi-scene or mixed ambiguity).**
>
> We extend our investigation of the perception-switching head mechanism to multi-scene ambiguity images. The Figure 7 in [Rebuttal PDF](https://anonymous.4open.science/r/ICLR_ambibench_anoymous-12A4/Ambi_Benchmark__ICLR_2026_Response_.pdf) indicates that existing a subset of perception-switching heads  corresponding to alternative interpretations on multi-scene images.
>
> Furthermore, we analyze the Pearson correlation of head-activation heatmaps for the same model across bistable and multi-scene categories. The results in following table show high consistency, further demonstrating the intrinsic nature of perception-switching heads, regardless of the data category.
>
> Table: Pearson Correlation of Heatmaps for Perception-Switching Heads (Bistable vs. Multi-Scene).
>
> | Model                     | Pearson Correlation |
> |---------------------------|----------------------|
> | Qwen2.5-VL-7B-Instruct    | 0.9134               |
> | Qwen2.5-VL-3B-Instruct    | 0.9025               |
> | InternVL3-8B              | 0.9338               |
> | InternVL3-2B              | 0.9652               |
>
> **Q3. The distinction from TET [1] can be further elaborated and discussed.**
>
> Thank you for pointing out this related work. In the revised manuscript, we have incorporated citations to [1].
> While TET primarily focuses on evaluating VLMs on low-level visual features and object-level perception, AmbiBench emphasizes higher-order cognition in ambiguous visual contexts. Specifically, AmbiBench includes multiple types of ambiguous images (bistable, hybrid, multi-scene, mixed) and evaluates VLMs across four complementary tasks (open-ended QA, multiple-choice, object localization, and local region description). This design allows AmbiBench to assess not only recognition accuracy but also reasoning, perceptual switching, and human-aligned cognitive processing.
>
> [1] PIXELS, PATTERNS, BUT NO POETRY: TO SEE THE WORLD LIKE HUMANS, 2025.
>
> **Q4. The experiments do not clearly specify the LLM judge’s criteria for verifying answers in open-ended and multiple-choice questions, particularly regarding the definition of all expected explanations. Providing additional details would enhance the reproducibility of the evaluation process.**
>
> We provide the LLM judge prompt along with several illustrative examples, which will be included in the revised manuscript.
>
> **LLM Judge Prompt:**
>
> You are a linguistic expert. Compare the following two sets of words and determine whether **every word** in Set 1 can be matched with **exactly one word** in Set 2 such that they meet **any** of these relationships:
>
> 1. They are **synonyms** (same or very similar meaning), e.g., "car" and "automobile".
>
> 2. They have a **hypernym/hyponym** relationship (one is a broader or narrower category of the other), e.g., "fruit" and "apple".
>
> 3. They are **exactly the same word**.
>
> Notes:
> - The order of words does not matter.
>
> - Each word in Set 1 must have a unique matching word in Set 2.
>
> - If **any word** in Set 1 cannot be matched, return: False.
>
> - Only if **all words** can be matched, return: True.
>
> Set 1: *predicted*
>
> Set 2: *ground_truth*
>
> Answer with only one word: True or False.
>
> **Examples**
>
> Predicted: Insect; Ground Truth: Insect; Response: True
>
> Predicted: Insect; Ground Truth: Spider; Response: True
>
> Predicted: Insect; Ground Truth: Leaves; Response: False

---

> ### Comment · Reviewer_CBZt · 2025-11-26
>
> Thank you for the authors' detailed response. I have carefully read the above content, which has resolved most of my questions. However, I still feel that the proportion of the mixed ambiguity category data for Q1 in the overall dataset is relatively small. Overall, I am maintaining my score.

---

> > ### Author Response · Authors · 2025-12-02
> >
> > Dear Reviewer,
> >
> > Thank you for your response, and we sincerely appreciate that most of your concerns have been resolved.
> >
> > Regarding your remaining point about the proportion of mixed ambiguity data, we have expanded the mixed ambiguity category to 40 images in the revised manuscript, making it comparable in size to the multi-scene category. The additional 12 mixed images can be found at [Rebuttal PDF](https://anonymous.4open.science/r/ICLR-rebuttal-2D57/Ambi_Benchmark__ICLR_2026_Response_.pdf). We have also updated all VLM results accordingly (Table 2 in Section 4.2). These revisions are highlighted in blue in the updated version.
> >
> > Sincerely,
> >
> > The Authors

---

### Official Review · Reviewer_Lgk3 · 2025-10-27

**Soundness:** 2
**Presentation:** 3
**Contribution:** 2
**Rating:** 2
**Confidence:** 3

**Summary:**

The paper introduces a new benchmark to investigate how VLMs perceive ambiguous images. They find that models perform worse than humans on all types of ambiguous images. They argue that humans can switch between local and global features, wheres VLMs are over-reliant on dominant features.

**Strengths:**

- *Originality*
The underlying question is interesting albeit a bit constrained.
- *Quality*
The idea behind the benchmark seems sensible, but I have doubts regarding the human experiment and reliability of the experiments.
- *Clarity*
The submission is clearly written.
- *Significance*
The paper introduces a new benchmark for ambiguous images, however the scope of the investigation is somewhat limited.

**Weaknesses:**

While the general question is interesting, there are a number of related studies investigating how aligned VLMs are with humans on specific perceptual tasks. While this paper adds some new data to this line of work, I don't think the study offers enough novel insights into the differences between human and machine vision, and it also does not provide solutions to bridge these differences.

**Questions:**

**Main questions:**
- This is very related to the literature on the perceptions of illusions in VLMs. I think this could be fleshed out a bit more in the related works section, including citations to [1].

- Can you give more examples of the Hybrid category? The example in the Appendix is not very obvious to me and I can not seem to see the lion in it. In fact, the example wrong VLM answer with "face, circle" seems more fitting to me than the example correct response.

- Human performance seems very low in the Hybrid and Multi-View experiments. In fact, it is so low that I think the task is too hard to really extract meaningful differences between models and humans. Additionally, you select a subset of 10 images per category, which is too little and you might end up selecting a subset that is a lot easier or harder on average compared to the full set. And then you only test 3 people on the 10 images if I understand correctly? That again is too little data to make a proper claim about human performance.

- If the number of humans tested and the number of images they are tested on really is as little as it seems, then I do not think the human performance reported here is reliable. Since parts of the stimuli in this paper are novel generated stimuli, we do not know how well they actually work as illusions (and looking at a few reported in this paper I am doubtful they are all meaningful). Human performance would be an important baseline to understand how well the stimuli work and if they are sensible illusions. Since this study does not offer a good measure of human performance it is very hard to put these stimuli and the model performance in context.

- I'd be really cautious about the take-aways from the Motion stimuli in the "Perception Catering to Human Preference" paragraph starting at line 425. Do you compare against negative examples here? As you hint yourself, models may have just learned that humans have this bias or they may be biased by the text prompt --- in any case, I would like to see a baseline here of what they answer for similar images that don't illicit this moving bias in humans.

- While the perceptual switch head analysis is interesting, I wonder what I should take away from it. Is the point that only the activation of a small number of heads changes between the two different interpretations of a given image? To me it is not entirely unsurprising that many heads do not show a large change, as at least the input image (and therefore likely the majority of input tokens) is still the same for both interpretations. I think to really get at a causal mechanism ("are these heads what allow for perceptual switching in these models") you would have to ablate the a top number of heads and see if the model now can't produce the alernative interpretation of the image anymore.

**Minor comments**:
- Caption for Figure 2: "categoreis" instead of categories
- Line 227 superfluous dot after subsection headline



[1] Ullman, Tomer. "The illusion-illusion: Vision language models see illusions where there are none." _arXiv preprint arXiv:2412.18613_ (2024).

---

> ### Author Response · Authors · 2025-11-21
>
> Dear Reviewer,
>
> We sincerely appreciate the time and effort you’ve invested in reviewing our paper. Below, we provide detailed responses to the
> questions you raised.
>
> **Novel insights compared to other related works.**
>
> 1. While some works have begun to evaluate the perceptual abilities
> of VLMs on ambiguous images, existing benchmarks are limited to a narrow set of categories, small in scale, and constrained in task design. A comparison can be found in Table 4 in the Appendix. This highlights the need for a comprehensive benchmark to assess how VLMs process and reason under perceptual ambiguity.
>
> 2. Unlike recent works that mainly focus on VLM reception, our study provides a mechanistic understanding of how VLMs process both perceptual and cognitive information. The capacity of VLMs to handle cognitive ambiguity is evaluated using bistable, multi-scene, and the newly introduced mixed categories. Importantly, we also explore perceptual-switch heads in VLMs and analyze their universal, sparse, and intrinsic properties across architectures.
>
> 3. Furthermore, we added new experiments to evaluate the functional roles of the identified perceptual-switch heads, demonstrating their importance and showing that positive interventions on these heads can further improve model performance. As shown in Table 1, masking the identified perceptual-switch heads leads to a substantial performance drop across all models, whereas masking an equal number of random heads results in only marginal degradation. Table 2 shows that enhancing the activation of these heads along their corresponding functional directions improves performance of VLMs on bistable and multi-scene images.
>
> Since managing ambiguity is a higher-order aspect of human intelligence, it relates to other visual abilities such as spatial intelligence (e.g., hybrid images involve distance-dependent perceptual ambiguity) and compositional reasoning (e.g., multi-scene images require a model to integrate both local and global information, identify objects across different contexts, and flexibly switch between two plausible scene interpretations). A comprehensive benchmark is therefore essential for systematically evaluating these capabilities and advancing broader visual understanding and reasoning.
>
> | Dataset | #Images | Ambiguity Types | Modalities | Task Types | Human Study |
> |---------|---------|----------------|------------|------------|------------|
> | HallusionBench | 72 | color, geometric | Image, video | multiple-choice | Yes |
> | Grounding-Illusions  | ~100 | color, geometric | image | open-ended, multiple-choice, localization | No |
> | SemVINK  | 112 | multi-scene (single hidden) | image | multiple-choice, open-ended | No |
> | IllusionVQA | ~400 | color, multi-scene(hidden), geometric | image | multiple-choice, open-ended, soft-localization | Yes |
> | IllusionBench+ | 1051 | geometric, bistable | images | multiple-choice, open-ended | Yes |
> | **AmbiBench (Ours)** | **>2000** | **9 Types (incl. Mixed)** | Image, Video | **multiple-choice, open-ended, localization, region-based description** | yes |
>
> Table 1. Negative Intervention of different VLMs on Bistable and Multi-Scene stimuli (score: LLM-Judge Accuracy).
>
> | Models | Inter_Head | Bistable | Multi-Scene |
> |-|-|-|--|
> | Qwen2.5-VL-3B-Instruct | random    | 38.18    | 55.26       |
> |                        | cognitive | 0.00     | 2.63        |
> | Qwen2.5-VL-7B-Instruct | random   | 41.82    | 55.26       |
> |                        | cognitive | 0.00     | 0.00        |
> | InternVL3-2B           | random   | 29.09    | 63.16       |
> |                        | cognitive | 0.00     | 0.00        |
> | InternVL3-8B           | random   | 38.18    | 63.16       |
> |                        | cognitive | 0.00     | 0.00        |
>
> Table 2. Positive Intervention of different VLMs on Bistable and Multi-Scene stimuli (score: LLM-Judge Accuracy).
>
> | Models                  | Inter_Head | Bistable | Multi-Scene |
> |-----|---|----|-----|
> | Qwen2.5-VL-3B-Instruct  | Before    | 43.64    | 57.89       |
> |                         | After     | **45.45**| **60.53**   |
> | Qwen2.5-VL-7B-Instruct  | Before    | 41.82    | 63.16       |
> |     | After     | **45.45**| **68.42**   |
> | InternVL3-2B            | Before    | 29.09    | 63.16       |
> |       | After     | **41.82**| **65.79**   |
> | InternVL3-8B            | Before    | 36.36    | 60.53       |
> |     | After     | **38.18**| 60.53       |
>
>
> **Q1. This is very related to the literature on the perceptions of illusions in VLMs. I think this could be fleshed out a bit more in the related works section, including citations to [1].**
>
> Thank you for pointing out this related work. In the revised manuscript, we have expanded the related works section to include a discussion of this line of research and incorporated citations to [1].
>
> [1] Ullman, Tomer. "The illusion-illusion: Vision language models see illusions where there are none." arXiv preprint arXiv:2412.18613 (2024).

---

> ### Author Response · Authors · 2025-11-21
>
> Thanks for your patience.
>
> **Q2. More examples of the Hybrid category**
>
> More hybrid examples can be found at [Rebuttal PDF](https://anonymous.4open.science/r/ICLR_ambibench_anoymous-12A4/Ambi_Benchmark__ICLR_2026_Response_.pdf). We also added clearer Hybrid example in the revised Appendix.
>
> **Q3. Human performance seems very low in the Hybrid and Multi-View experiments.**
>
> 1. **Human evaluation on ambiguous images has substantial variability and is difficult to fully control**. The lower accuracy in Hybrid and Multi-View categories likely reflects the viewing-condition requirements of these stimuli: hybrid images often require stepping back or reducing resolution to perceive the second interpretation, while multi-view images require rotating the picture or tilting one's head. Although we instructed annotators about these conditions, **the online crowdsourcing setup prevents us from controlling viewing distance or head orientation**. Despite this, the **gap between humans and VLMs remains meaningful**, highlighting intrinsic differences in how the two systems process ambiguous information.
>
> 2. While we selected 10 images per category for the online human test, each image in **our dataset had already been validated by three independent annotators with 100\% agreement (lines 245–248 of the paper).** The additional 10-image subset was used only to mitigate annotator bias in the behavioral comparison, and the images were randomly selected.
> Moreover, the central objective of the paper is to analyze VLM behavior, for which we evaluated 12 state-of-the-art VLMs across the full dataset.
>
> **Q4. If the number of humans tested and the number of images they are tested on really is as little as it seems, then I do not think the human performance reported here is reliable.**
>
> As described in Section 3.2.4, we use a rigorous two-stage filtering and annotation pipeline. Every image in AmbiBench is reviewed by three independent annotators, and **only images with 100\% agreement are retained.** The lower scores observed in the Hybrid and Multi-View categories are discussed in Q2 (we will add these analysis in the new version.) Importantly, this does not undermine the validity of the illusions themselves, which were already verified through the earlier annotator stage.
>
> Furthermore, we emphasize our **strict data filtering** with following aspects: 1. For categories such as bistable, we rely on well-established illusions from online sources and prior literature [1]. Out of 203 samples in [1], only 2 remained after our strict filtering protocol.
> 2. For categories such as hybrid, we use high-quality AI generation methods [2]. With strict filtering, only 277 hybrid images are retained in Ambibench from over 2000 images.
>
> [1] Newen, Carina, et al. "Do you see what I see? An Ambiguous Optical Illusion Dataset exposing limitations of Explainable AI." arXiv preprint arXiv:2505.21589 (2025).
>
> [2] Geng, Daniel, Inbum Park, and Andrew Owens. "Visual anagrams: Generating multi-view optical illusions with diffusion models." Proceedings of the IEEE/CVF Conference on Computer Vision and Pattern Recognition. 2024.
>
> **Q5. I'd be really cautious about the take-aways from the Motion stimuli in the "Perception Catering to Human Preference" paragraph starting at line 425. More baselines.**
>
> We constructed a baseline set consisting of 44 images from the COCO dataset—matching the number of motion images—whose labels indicate no motion. As shown in the table below, most VLMs can accurately label these static images, while some models (e.g., Phi-4-multimodal-instruct) still achieve only around 50\% accuracy, essentially performing at chance level.
>
> Furthermore, when we invert the prompt and ask whether the motion images in AmbiBench are static, some models exhibit a substantial performance drop. This asymmetry indicates that the models' responses are highly sensitive to the phrasing of the human query, suggesting that their behavior is at least partially driven by catering to human preference or expectation.
>
> Overall, the motion ambiguity images remain challenging for existing VLMs. We will add these analysis in the new version.
>
> Table. Performance of 12 VLMs and humans on COCO images, Motion (static prompt), and Motion (ours).
>
> | Model  | COCO image | Motion (static prompt) | Motion (ours) |
> |-|-|--|-|
> | Gemini 2.5 Pro  | 96.00     | 16.00  | 76.00  |
> | GPT5  | 100.00    | 0.00    | 48.00   |
> | Qwen2.5-VL-7B-Instruct   | 100.00 | 68.00    | 0.00   |
> | Qwen2.5-VL-72B-Instruct  | 100.00    | 76.00    | 36.00  |
> | InternVL3-8B-Instruct   | 100.00    | 8.00    | 8.00   |
> | InternVL3-78B-Instruct    | 100.00    | 72.00   | 44.00  |
> | llava-hf/llava-1.5-13b-hf    | 64.00     | 36.00   | 96.00  |
> | Kimi-VL-A3B-Thinking-2506  | 100.00    | 12.00   | 40.00  |
> | Phi-4-multimodal-instruct  | 52.00     | 12.00  | 72.00   |
> | Idefics3-8B-Llama3   | 96.00  | 24.00| 8.00   |
> | Gemma-3-27b-it   | 64.00 | 0.00  | 76.00  |
> | Ovis2-34B  | 100.00  | 32.00  | 60.00  |

---

> ### Author Response · Authors · 2025-11-21
>
> Thanks for your patience.
>
> **Q6. Causal evidence of perceptual-switch heads.**
>
> We conducted negative intervention experiments by masking the activations of the identified perceptual-switch heads. As shown in Table 1 (in response to “Novel insights compared to other related works”), the models’ ability to generate alternative interpretations drops significantly—sometimes even to zero—indicating that these heads play a causal role in enabling perceptual switching.
>
> **For other mirror comments**
>
> Thanks for pointing these out.
> We have corrected the typo in the caption of Figure 2 (“categoreis” → “categories”) and removed the superfluous dot after the subsection headline in line 227.

---

> ### Comment · Reviewer_Lgk3 · 2025-11-25
> **Reviewer response to Author comments**
>
> Dear Authors,
>
> Overall, I appreciate the extensive rebuttal response and that you have run additional analyses for Q5 and a more functional analyses of the perceptual switch heads. I have decided to raise my score, but retain worries about parts of the benchmark that make me unable to recommend clear acceptance at this point. Below I will outline my remaining concerns:
>
> My concerns revolve around points 2-4 of the main questions in my initial review: The Hybrid stimuli are not entirely obvious to me and even in the new examples you offered in the rebuttal I am struggling to see both categories. To add on to this, Human performance seems very low in the Hybrid and Multi-View experiments --- you yourself write that testing humans on some of these is hard in an online setting. Since these are novel stimuli, and the accuracies are so low, I am unsure if the task really features meaningful ambiguous images. Furthermore, I retain my concerns about the number of humans tested and the number of images they are tested. Especially, since parts of the stimuli in this paper are novel generated stimuli, we do not know how well they actually work as illusions. Human performance would be an important baseline to understand how well the stimuli work and if they are sensible illusions. Since this study does not offer a good measure of human performance it is very hard to put these stimuli and the model performance in context.

---

> ### Author Response · Authors · 2025-11-27
>
> Dear Reviewer,
>
> Thank you very much for your thoughtful follow-up review and for raising your score. We sincerely appreciate the time you invested in evaluating our work. We address your remaining concerns below.
>
> **1. Newly generated stimuli show similar performance to stimuli from online sources for humans.**
>
> For hybrid images in AmbiBench, 29 images were selected from online sources. A general hybrid image sourced online was used as a test example to ensure the reliability of the human study—participants’ responses were deemed valid only if answered correctly. We also randomly selected the same number of newly generated hybrid images. The same three participants evaluated both subsets of hybrid images. As shown in the table below, the mean human accuracy across the three participants was 67.82\% for Hybrid (online source) and 63.22\% for Hybrid (newly generated). This indicates that **our novel stimuli generated using diffusion models are valid**. The examples of hybrid images at [Rebuttal PDF](https://anonymous.4open.science/r/ICLR_ambibench_anoymous-12A4/Ambi_Benchmark__ICLR_2026_Response_.pdf) are drawn from both online sources and newly generated images.
>
> **2. Meaningful ambiguous images also differ across individuals.**
>
> In addition, we note that human performance exhibits substantial variability across participants. For example, among the three participants, one may achieve over 80\% accuracy while another achieves around 60\%. Ambiguous images that rely on abstract interpretation or imagination remain challenging for humans, and differences in background and prior knowledge among individuals contribute to this variability. A classic example is The Dress (2015), in which different observers perceived different colors under the same viewing conditions, illustrating that even well-controlled stimuli can elicit divergent perceptual experiences.
>
> For the multi-view category, we selected 30 images, and human performance is reported in the following table.
>
> Notably, AmbiBench has already been validated by three independent annotators (Lines 265–266 in the revised version). Using three human evaluators is consistent with common evaluation protocols adopted in prior work [1].
>
> Table. Human Performance on more hybrid and multi-view images.
> | Human       | Hybrid (online source) | Hybrid (newly generated) | Multi-view |
> | ----------- | ---------------------- | ---------------------- | ---------- |
> | 1           | 82.75                  | 58.62                  | 68.97      |
> | 2           | 58.62                  | 65.62                  | 41.38      |
> | 3           | 62.06                  | 65.62                  | 44.83      |
> | **Average** | 67.82                  | 63.22                  | 51.72      |
>
> We will update the manuscript based on our new results that incorporate more examples. Moreover, the main objective is AI, i.e., **“What do VLMs see?”**. AmbiBench employs **a strict two-stage annotation**, and **the gap between humans and VLMs remains meaningful**. Totally, AmbiBench provides a comprehensive benchmark for assessing how VLMs process and reason under perceptual ambiguity.
>
> We hope this adequately addresses your concerns. We are happy to engage in further discussion if you have any additional suggestions.
>
> [1] Shahgir, Haz Sameen, et al. "Illusionvqa: A challenging optical illusion dataset for vision language models." COLM, 2024.
>
>
> Best Regards,
>
> The Authors

---

### Author Response · Authors · 2025-12-01
**Summary of Updates**

Dear AC and SAC,

We would like to express our sincere gratitude for your handling of our paper, "What Do VLMs See? Benchmarking Vision-Language Models on Ambiguous Images", and for the constructive feedback and suggestions provided by the reviewers. During the rebuttal phase, we received follow-up responses from **Reviewer Lgk3 and Reviewer KZ8j** (updated positive scores), both of whom acknowledged our clarifications and raised their evaluations.

We summarize the **main contributions of this work** below:

- We present AmbiBench, the first systematically curated benchmark for evaluating ambiguous image perception in VLMs. It spans nine categories of visual ambiguity—including a newly introduced mixed category—and supports text, image, and video modalities.

- We evaluate 12 state-of-the-art VLMs across open-ended, multiple-choice, ambiguous localization, and local-region description tasks, directly comparing model outputs with human responses and revealing substantial differences in ambiguity handling.

- We explore perceptual-switch heads in VLMs and analyze their universal, sparse, and intrinsic properties across architectures. We further examine their functional roles and demonstrate that positive interventions improve performance, highlighting their importance in resolving perceptual ambiguity.

We have carefully addressed all reviewer concerns and revised the manuscript accordingly.
Below we summarize the updates made in the revised manuscript (all changes highlighted in blue):

**For Reviewers Lgk3,  GvwZ, and KZ8j:**
Added new experiments, including **negative interventions demonstrating the functional role of perceptual-switch heads and positive interventions showing performance improvements** (lines 498–526, Section 4.4).
(Both Reviewer Lgk3 and KZ8j acknowledged the value of these new insights.)

**For Reviewers Lgk3 and CBZt:**
Added related work (lines 45, 142–143).

**For Reviewers GvwZ and KZ8j:**
Added **a comparison between AmbiBench and other benchmarks** for color ambiguity (lines 447–450, with details in Table 13 of Appendix A.6).

**For Reviewer Lgk3:**
Included **new hybrid-image example** (line 1596, Appendix A.10), **updated human results on hybrid and multi-view images** (Table 1, Section 4.2, lines 1074-1101 in Appendix A.4), added **a new analysis on motion ambiguity** (lines 429–430, Appendix A.7), and corrected typographical errors.

**For Reviewer CBZt:**
**Expanded the mixed category to 40 images (match the size of the multi-scene category)**, and updated VLM results accordingly (Table 2, Section 4.2). Added analysis of perceptual-switch-head mechanisms on multi-scene images (lines 482–485, Appendix A.9). Added details of the LLM-Judge setup (Appendix A.5).

**For Reviewer GvwZ:**
Expanded the discussion on how ambiguous-image tasks relate to broader visual reasoning and compositional understanding (lines 536–538, Conclusion).
(The importance of perceptual-switch heads and ways to reduce the VLM–human gap are addressed through the new negative and positive intervention experiments.)

**For Reviewer KZ8j:**
Added a discussion of AmbiBench’s limitations, including category imbalance (line 539, Conclusion). Added statistical analyses for robustness and visualization analyses (lines 451–452, with details in Appendix A.6 and A.8).

Thank you very much for your time and thoughtful consideration.

Best regards,

The Authors

---

### Meta-Review · Area_Chair_VwpD · 2026-01-05

**Summary:**

The key reviewer concerns are:

* Distinctiveness and novelty vs. prior illusion benchmarks; need for clearer value and cross-benchmark positioning [GvwZ,KZ8j]

* Human study reliability and scope: small N, online viewing constraints; validity of Hybrid and Multi-View categories [Lgk3]

* Dataset imbalance and scale: particularly the small Mixed category; implications for statistical power [CBZt,KZ8j]

* Mechanistic claims and causality: perceptual-switch heads need causal evidence beyond correlation effects [Lgk3,GvwZ,KZ8j]

* Motion stimuli and phrasing bias: need baselines and inverted prompts to rule out prompt-driven effects [Lgk3]

* Statistical robustness, LLM-judge criteria, and multiple-testing corrections [KZ8j,CBZt]

* Data contamination: reliance on online sources; need for mitigation and calibration [GvwZ]

As outlined below, the authors provided a constructive rebuttal with new experiments (interventions, expanded Mixed category, statistical robustness, motion baselines, benchmark comparisons). While this addresses several points, core concerns about human-study reliability / Hybrid-clarity, distinctiveness beyond aggregation, and methodological limitations remain. I believe the submission would benefit from a more thorough resolution of the remaining concerns and an additional review cycle. I propose reject.

**Reviewer Concerns:**

* Positioning and comparisons: authors added valid cross-benchmark results and attempted to link ambiguous-image competence to broader reasoning abilities.

*  Mechanistic evidence: authors performed negative interventions, causing near-complete performance collapse on bistable and multi-scene, and positive interventions that improved accuracy, supporting causal relevance.

* Prompt sensitivity: controls and inverted prompts revealed asymmetries and phrasing sensitivity, towards strengthening the 'preference-catering' analysis.

*  Statistical robustness and LLM-judge clarity: multi-seed runs, significance tests with multiple-testing corrections and LLM-judge prompts were provided.

* Contamination mitigation: authors emphasised generation of new ambiguous images and task designs; however, they concede that a definitive audit is not feasible and a cross-model contamination risk remains.

In particular, several raised concerns can be considered to remain:

*  Human study reliability remains limited by small N and uncontrolled viewing conditions; despite image validation and added examples, reviewers still question interpretability of low human scores and the clarity of the test images. I tend to agree that this raises an important point.

*  Distinctiveness vs. prior work: although comparisons and mechanistic analyses help, some reviewers remain unconvinced that novelty goes beyond careful aggregation plus interpretability, especially given overlaps with existing work.

*  Dataset imbalance and scope: Even with moderate expansion, category imbalance persists; implications are acknowledged but not fully resolved.

*  Broader methodological breadth: visualisations rely on attention; other mechanistic tools (e.g., SAEs) were suggested but not incorporated; some statistical critique appears to remain regarding depth of robustness but the discussion was likely cut short. On balance this point can be considered somewhat resolved.

*  Authors concede that no definitive contamination audit is possible; reliance on online sources persists for part of the data.

**Reviewer Scores:**

* Reviewer Lgk3: initially critical on (Hybrid) clarity, human baselines, and a need for causal head evidence. Updated head ablations / interventions, motion baselines likely improved the assessment; the reviewer reported somewhat raising their score but retained concerns about Hybrid stimuli and human testing and was explicitly unable to yet champion the submission.

* Reviewer CBZt: raised concerns regarding Mixed size and generalisation of switch-heads. Authors expanded Mixed and added cross-category head analyses and judge details; reviewer maintained their marginally positive score.

* Reviewer GvwZ: Questioned distinctiveness, causality, and contamination handling. Despite added comparisons and interventions, I feel that core positioning concerns and causality reservations likely largely remained.

* Reviewer KZ8j: raised issues on novelty, AI-generated image quality, imbalance, comparisons, and statistical rigor. Authors added cross-benchmark comparisons, statistical robustness with multi-seed runs and multiple-testing corrections, and visualization methodology. The reviewer reported upgrading their score but still flagged multiple-testing considerations (addressed subsequently by authors).

Decision recommendation

This submission presents a carefully engineered benchmark with interesting mechanistic follow-up and (latterly improved) statistical reporting. However, I note that key reservations persist: (i) lingering questions about human-study reliability and Hybrid clarity, (ii) incremental distinctiveness relative to existing benchmarks despite added comparisons, and (iii) category imbalance and methodological breadth. I feel that thorough resolution of the remaining concerns would materially strengthen the case for acceptance in a future submission.

---

### Decision · Program_Chairs · 2026-01-26

Reject